# Elevated temperature and CO$_2$ strongly affect the growth strategies of soil bacteria

Yang Ruan[1,2], Yakov Kuzyakov[3,4], Xiaoyu Liu[2], Xuhui Zhang[2], Qicheng Xu[2], Junjie Guo[2], Shiwei Guo[2], Qirong Shen [2], Yunfeng Yang [5] & Ning Ling [1,2] ✉

The trait-based strategies of microorganisms appear to be phylogenetically conserved, but acclimation to climate change may complicate the scenario. To study the roles of phylogeny and environment on bacterial responses to sudden moisture increases, we determine bacterial population-specific growth rates by $^{18}$O-DNA quantitative stable isotope probing ($^{18}$O-qSIP) in soils subjected to a free-air CO$_2$ enrichment (FACE) combined with warming. We find that three growth strategies of bacterial taxa – rapid, intermediate and slow responders, defined by the timing of the peak growth rates – are phylogenetically conserved, even at the sub-phylum level. For example, members of class Bacilli and Sphingobacteriia are mainly rapid responders. Climate regimes, however, modify the growth strategies of over 90% of species, partly confounding the initial phylogenetic pattern. The growth of rapid bacterial responders is more influenced by phylogeny, whereas the variance for slow responders is primarily explained by environmental conditions. Overall, these results highlight the role of phylogenetic and environmental constraints in understanding and predicting the growth strategies of soil microorganisms under global change scenarios.

Increasing emissions of greenhouse gases intensify global warming[1–3], affecting belowground processes directly and indirectly[1,2,4]. Community composition and functions of soil microbes, the primary mediators of belowground processes, can be modified by long-term climate change[1]. This slow acclimation, however, disregards the response of decomposers to sudden events, e.g., nutrient pulses by rewetting dry soil, which strongly impacts biogeochemical processes[5–7]. In agroecosystems, wet-up events (such as irrigation or precipitation) cause acute changes in water potential. The sudden increase of water availability after rewetting of dry soil dissolves organic compounds, resulting in a nutrient pulse[6,8,9]. At the same time, the sharp rise of osmotic potential induces microbial cell burst and efflux of intracellular solutes, which is a significant C and nutrient source for the survivors[8]. The transient nutrient pulse mediated by wet-up events triggers microbial

activity and reproduction, further stimulating organic matter mineralization[8–10]. Nonetheless, a knowledge gap remains about the patterns of microbial growth response to soil wet-up under climate change scenarios.

Soil microbiota respond to C and nutrient pulses with distinct strategies: Some taxa, featuring a rapid metabolic response, recover within 1 h of wet-up, whereas other species have a long lag-phase and achieve maximum growth rates after one week of wet-up, yet still efficiently compete for resources[6,8,11]. These physiological traits rely on a high degree of gene coordination and have non-random phylogenetic patterns[11,12]. Microbial phylogeny offers a framework to clarify the response mechanisms to specific environments and, consequently, the particular functions of microbial taxa within an ecosystem[11,13]. Microbial phylogeny is directly related

[1]Centre for Grassland Microbiome, State Key Laboratory of Herbage Improvement and Grassland Agro-Ecosystems, College of Pastoral Agriculture Science and Technology, Lanzhou University, Lanzhou 730020 Gansu, China. [2]Jiangsu Collaborative Innovation Center for Solid Organic Waste Resource Utilization, Nanjing Agricultural University, Nanjing 210095, China. [3]Department of Soil Science of Temperate Ecosystems, Department of Agricultural Soil Science, University of Goettingen, Göttingen 37077, Germany. [4]Peoples Friendship University of Russia (RUDN University), 117198 Moscow, Russia. [5]State Key Joint Laboratory of Environment Simulation and Pollution Control, School of Environment, Tsinghua University, Beijing, China. ✉e-mail: nling@njau.edu.cn

to its phenotypic and functional traits. These traits include pH and salinity tolerance[13], the utilization of various organic substrates[14], and the metabolic as well as growth response to pulse events[8,15], with consequences for the overall community assembly. We therefore first tested the hypothesis that the microbial growth pattern in response to nutrient pulses following wetting events is phylogenetically conserved.

Microbes can shift their growth strategies to acclimate to changed environments[5,16,17]. Microbial responses to water pulses after drought depend on multiple environmental and biotic factors such as the presence of labile organic matter[18–20], the conditions in soil microhabitats[8,21], and the interactions among microbial populations[5,22]. All these can be greatly modified by long-term climate change. Indeed, precipitation history has influenced the strategies of individual microbial groups, which are inconsistent with the behavior based on their phylogenetic descent[5]. Nonetheless, how long-term climate changes shape microbial growth strategies after soil wet-up and influence their phylogenetic structures remain largely unknown. We traced microbial growth strategies by measuring growth dynamics at the population level after rewetting dry soil and tested the second hypothesis that long-term warming and elevated $CO_2$ concentrations shift the pre-existing growth strategies of certain species, which reduces phylogenetic coherence.

The eco-physiology of microbes is generally expected to be phylogenetically conserved because of the evolutionary history of a given species (i.e., its "congenital endowment")[12]. At the same time, environmental conditions (i.e., the "acquired environment" of a given species) also shapes microbial functions (e.g., growth) and genome architecture[16,23]. Accordingly, the ecological traits of microorganisms mirror both phylogenetic constraints and environmental acclimation. However, the degree to which these traits are affected by phylogeny and environment remain uncertain.

Here, we performed [18]O-quantitative stable isotope probing ([18]O-qSIP) to characterize bacterial growth after the soil planted with rice and wheat was exposed to a decade of warming and $CO_2$ enrichment (Fig. 1a). Taxon-specific population growth rates were assessed in three time intervals by [18]O incorporation within DNA molecules[24]. Three species growth strategies—rapid, intermediate, and slow response—were categorized based on the timing of the maximal growth rates (Fig. 1d). We identified the phylogenetic patterns of these three growth strategies as affected by warming and $CO_2$ enrichment. The reshaping of the growth strategies and their phylogenetic patterns enabled us to quantify the specific contributions of phylogenetic constraints and environmental acclimation to microbial growth.

## Results

The Free Air Carbon dioxide Enrichment (FACE) experiment was established in 2010 with four simulated climate change scenarios: ambient temperature and $CO_2$ concentration (Contr), elevated temperature (= warming) of canopy air by +2 °C (eT), elevated $CO_2$ concentration up to 500 ppm (eCO$_2$), and combined $CO_2$ enrichment and warming (eTeCO$_2$) (Fig. 1b). Warming and elevated $CO_2$ altered plant physiological traits: Total biomass, grain yield, and gross photosynthetic rate decreased under warming but increased under elevated $CO_2$ concentration[25,26]. Soil pH decreased, but organic carbon content increased in all three climate scenarios[27]. Based on additional soil sampling (in July 2022), we confirmed the pulses of available organic substances and nutrients to microbes after rewetting of dry soils, indicated by increased $CO_2$ fluxes and fluorescein diacetate (FDA) hydrolase activity (Supplementary Fig. 1). The dissolved organic carbon was exhausted after one day but dissolved nitrogen increased gradually during the 6-d incubation, pointing to the decomposition of accessible and labile organic matter and nitrogen accumulation (Supplementary Fig. 1).

### Growth responses of active bacteria as affected by climate change

Quantitative [18]O stable isotope probing was used to calculate growth rates of individual species with high accuracy and specificity[24]. Soil samples for qSIP incubation were collected in June 2020, i.e., ~10 years after the start of climate simulation experiment. The unfractionated DNA (i.e., DNA extracted from soils sampled immediately after wet-up) was examined by 16S rRNA amplicon sequencing. The bacterial community in eT had the lowest richness, whereas the elevated eCO$_2$ concentration increased α-diversity compared with Contr ($p < 0.05$, Supplementary Fig. 2a). Despite the similar community composition at the phylum level (Supplementary Fig. 2b), the OTU abundances were clearly separated between climate change treatments (PERMANOVA, $R^2 = 0.618$, $p = 0.001$, Supplementary Fig. 2c). Consequently, the community composition of soil microbes has changed and acclimated to a decade of warming and elevated $CO_2$ in the wheat-rice rotation system.

Excess atom fraction [18]O values (EAF, Supplementary Fig. 3) and the population growth rate of each OTU was calculated using the qSIP pipeline (see "Methods"). Collectively, 1017 OTUs were identified as [18]O "incorporators" (i.e., OTUs with growth rates significantly greater than zero) and used for subsequent growth assessment (Supplementary Data 1). Cumulative growth rates for the whole communities were calculated over each of the three incubation periods (0–1, 0–3, and 0–6 d incubation, Fig. 2), according to a previous publication[6]. Both the duration after rewetting and climate influenced microbial growth rates (both $p < 0.001$, two-way ANOVA). During the 6-d incubation after rewetting, the community-level growth rates peaked within the first day in all four climate change scenarios (no significant difference between 0–1 d and 0–3 d incubations in the eCO$_2$ treatment). The growth rates decreased over time, and declined by 73%, 87%, 75%, and 50% at the 6-d incubation relative to the first day of incubation in the Contr, eT, eCO$_2$, and eTeCO$_2$ soils, respectively. Among the four climate conditions, the peak community-level growth rates under eT and eCO$_2$ maintained levels equivalent to Contr, but declined by 47% in eTeCO$_2$ ($p < 0.05$).

Taxa with relatively high growth rates belonged to several bacterial phyla (Fig. 2 and Supplementary Fig. 4), including Actinobacteria (average relative growth rate: 29% of new 16S rRNA gene copies), Firmicutes, Bacteroidetes, Acidobacteria, and Gammaproteobacteria (14%, 13%, 11%, and 9%, respectively). Some phyla had similar growth dynamics (i.e., changes in growth rates along the incubation time) among climate change scenarios (Supplementary Fig. 4). For example, the growth rates peaked on day 1 for Gammaproteobacteria, Bacteroidetes, and Actinobacteria, but on day 3 for Deltaproteobacteria and Chloroflexi in all climate change scenarios. The consistent growth trajectories demonstrate that the growth responses of such phyla could be robust and predictable in future climate change. In contrast, the time points at which growth rates peaked for several phyla such as Acidobacteria and Firmicutes were dependent on climate scenarios, indicating direct climate sensitivity or indirect: due to changes of plant rhizodeposition.

There was no correlation between the per-capita growth rates and initial population density in the control soil (line model: $p > 0.05$). By contrast, we recorded significant negative correlations under eT and eCO$_2$ scenarios ($R^2 = 0.037$ in eT, and 0.062 in eCO$_2$; $p < 0.01$, Supplementary Fig. 5), indicating negative density-dependent selection[6,28]. The negative relationships, however, disappeared by the combined manipulations of $CO_2$ and temperature ($p > 0.05$).

### Bacterial growth strategies are affected by phylogeny

We classified bacterial species into three growth strategies: rapid responders (OTUs with peak average growth rates on ~1 d), intermediate responders (OTUs with peak rates on ~3 d), and slow responders (OTUs with peak rates on ~6 d). A phylogenetic tree including all responders under four climate conditions was

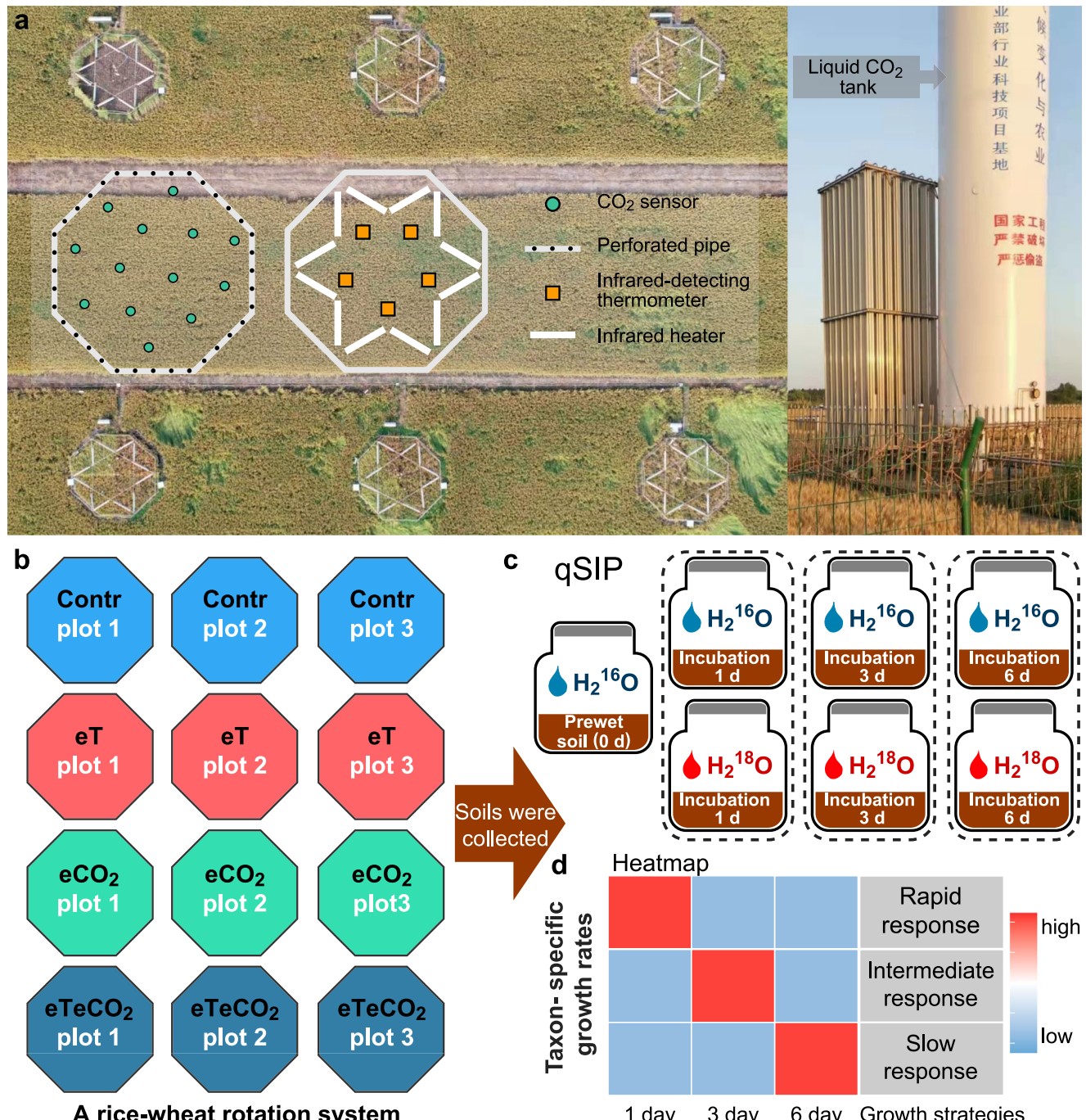

**Fig. 1 | Field and laboratory experimental design.** To examine the effects of $CO_2$ enrichment (500 ppm $CO_2$) and warming (+2 °C) for a typical rice-wheat rotation system, an open-air experimental field was started in 2010 (**a**). Four treatments with twelve 50 $m^2$ rings (each treatment with three biological replicates) were set (**b**). A qSIP incubation experiment was performed to determine the growth strategies of active microbes in each climate change treatment (**c**). Briefly, 2 g air-dried soils with 400 μL of natural abundance water ($H_2^{16}O$) or 98 atom% $H_2^{18}O$ were incubated in the dark at room temperature, with harvests at 0 day, 1 day, 3 days, and 6 days. We calculated the taxon-specific growth rates and determined which of the three pre-defined growth strategies an OTU exhibited in each simulated climate change treatment (**d**). Contr represents the ambient temperature and $CO_2$ concentration; eT represents warming only; $eCO_2$ represents elevated $CO_2$ concentration only; and $eTeCO_2$ represents combined warming and $CO_2$ enrichment.

constructed, and three phylogenetic indices were used to estimate the distribution patterns of three growth strategies along the given phylogeny. First, phylogenetic dispersion (D) was calculated to compare whether the observed phylogenetic pattern of traits was closer to random or Brownian motion patterns[12,29]. A smaller D value indicated a higher degree of phylogenetic clustering. All $p$ values of phylogenetic dispersion (D) in the three growth strategies suggested phylogenetically nonrandom distributions ($p < 0.001$, Table 1). D values were the lowest for rapid responders (0.38) and the highest for slow responders (0.77), clearly reflecting the reduction of phylogenetic convergence from rapid to slow growth strategies.

We used two additional indices, i.e., Blomberg's K and Pagel's λ, to test for significantly nonrandom phylogenetic distributions (i.e., phylogenetic signal)[23,30]. In most cases, the range of Pagel's λ was 0–1 (from random distribution to strong phylogenetic signal), whereas Blomberg's K can be higher than 1, indicating stronger trait

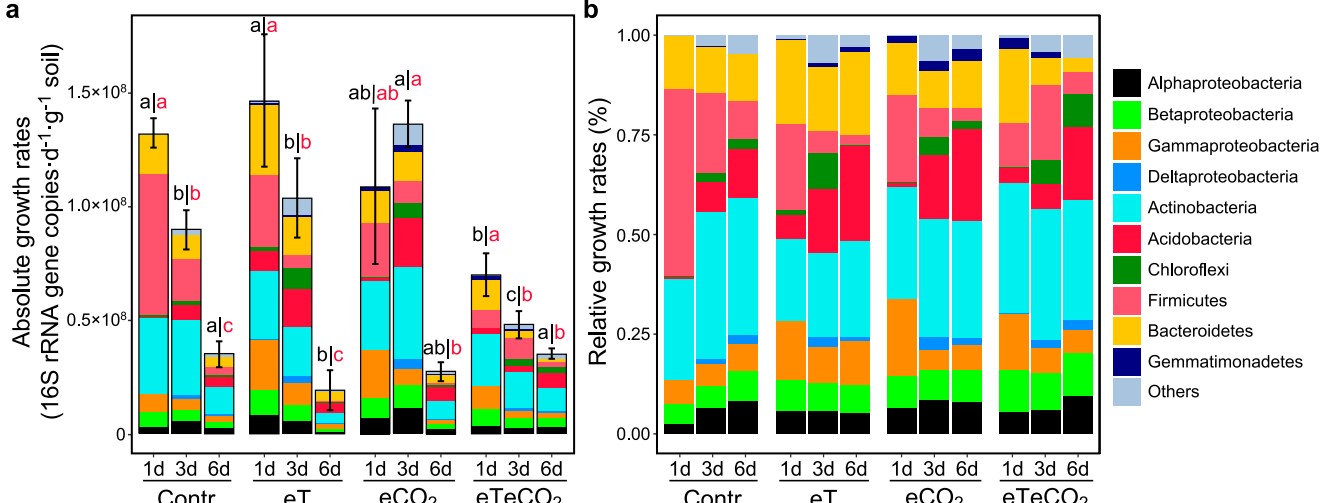

**Fig. 2 | Growth dynamics of the bacterial communities in soils under different climate change scenarios after water addition.** Absolute (**a**) and relative population growth rates (**b**) of bacterial phyla. Values were averaged across replicates ($n = 3$ biologically independent samples). Error bars: standard deviation (SD). The proportion of growth rate for each phylum (i.e., relative growth rate) was relativized by the total growth rate of the treatment. Bar color: bacterial phylum or class (10 of them are shown, accounting for > 93% of growth rates, and remaining phyla are designated as "Other"). ANOVA with two-sided LSD multiple comparisons was used for the statistical analysis. Letters: significant differences between treatments ($p < 0.05$). Black letters (left): significant differences among climate change scenarios at the same time point; Red letters (right): significant differences among time points within each climate treatment. Contr represents the ambient temperature and $CO_2$ concentration; eT represents warming only; $eCO_2$ represents elevated $CO_2$ concentration only; and $eTeCO_2$ represents combined warming and $CO_2$ enrichment.

## Table 1 | Phylogenetic indices of three growth strategies

|  | Phylogenetic dispersion | | Blomberg's K | | Pagel's λ | | Nearest taxon index | |
|---|---|---|---|---|---|---|---|---|
|  | D | p value | K | p value | lambda | p value | NTI | p value |
| Rapid | 0.38 | <0.001 | 0.16 | 0.001 | 0.59 | <0.001 | 3.18 | 0.001 |
| Intermediate | 0.59 | <0.001 | 0.12 | 0.003 | 0.40 | <0.001 | 0.06 | 0.466 |
| Slow | 0.77 | <0.001 | 0.11 | 0.002 | 0.24 | <0.001 | 4.11 | 0.001 |

The smaller D values represent more clustered phylogenetic patterns, while the smaller K and λ indicate more random phylogenetic patterns. A permutation test was used for data analysis and the *p* values were calculated with 1000 permutations based on trait prevalence and phylogeny.

similarity between related species. Similar to phylogenetic dispersion (D), the K and λ showed phylogenetic signals of distribution patterns in three growth strategies ($p < 0.01$); their strength gradually weakened from rapid to slow growth strategies (Table 1). The nearest taxon index (NTI) showed a clustered distribution of rapid and slow responders at the end of the branches of the phylogenetic tree. Collectively, bacterial growth responses to sudden moisture increase are influenced by phylogeny, suggesting that under the conditions evaluated here, vertical inheritance was essential for microbial growth traits.

**Environmental acclimation caused by climate change reshapes microbial growth and partly modifies the phylogenetic patterns**
The proportion of active microbes with the three growth strategies depended on climate history (Fig. 3a). In the control soil, 14% of OTUs were identified as rapid responders, and 61% and 25% of active taxa were intermediate and slow responders, respectively. After a decade of acclimation to warming-only or $CO_2$ enrichment-only, the proportion of rapid and intermediate responders increased (rapid: 21% and 18%, intermediate: 77% and 80% in eT and $eCO_2$ treatments, respectively), while only 2% of OTUs were identified as slow responders. An opposite trend, however, was recorded in the soil subjected to combined warming and $CO_2$ enrichment: The proportion of slow responders was the highest among the four climate scenarios, increasing by 164% relative to the control soil.

Climate change shifted growth strategies at the OTU level (i.e., the same OTU was assigned into distinct strategies when comparing

control soil with the other three climate change treatments, Fig. 3b). To explore the response of microbial growth strategies to warming and $CO_2$ enrichment, two shift types were defined: growth accelerated and growth delayed. "Growth accelerated" meant the maximum growth rates came earlier after experiencing climate change (Fig. 4a). Conversely, "growth delayed" meant the maximum growth rates were delayed after experiencing climate change (Fig. 4b). A total of 224 OTUs, mainly from Alphaproteobacteria, Actinobacteria, and Acidobacteria, were detected as growth accelerated after warming, whereas $CO_2$ enrichment accelerated the growth of 217 OTUs. However, only 74 OTUs were detected as "growth accelerated" when comparing the growth strategies in control soil with that in $eTeCO_2$ (Fig. 4a). At the same time, 28 OTUs and 15 OTUs were "growth delayed" when comparing control soil with eT and $eCO_2$ (Fig. 4b), respectively. Importantly, the number of "growth delayed" OTUs increased by nearly 6 times after experiencing warming and $CO_2$ enrichment simultaneously. Collectively, those microbial species acclimated to long-term warming-only or elevated $CO_2$-only responded faster to the wet-up events than those under ambient conditions. In contrast, the microbes in soil under the combination of warming and $CO_2$ enrichment tended to slow their growth response.

Among the taxa with changed growth strategies, a total of 45 OTUs were shared among the three treatment comparisons (i.e., Contr vs eT, Contr vs $eCO_2$, and Contr vs $eTeCO_2$), including 42 OTUs with "growth accelerated" and 3 with "growth delayed" OTUs (Fig. 4a, b). Based on the visualized phylogenetic distribution of these OTUs

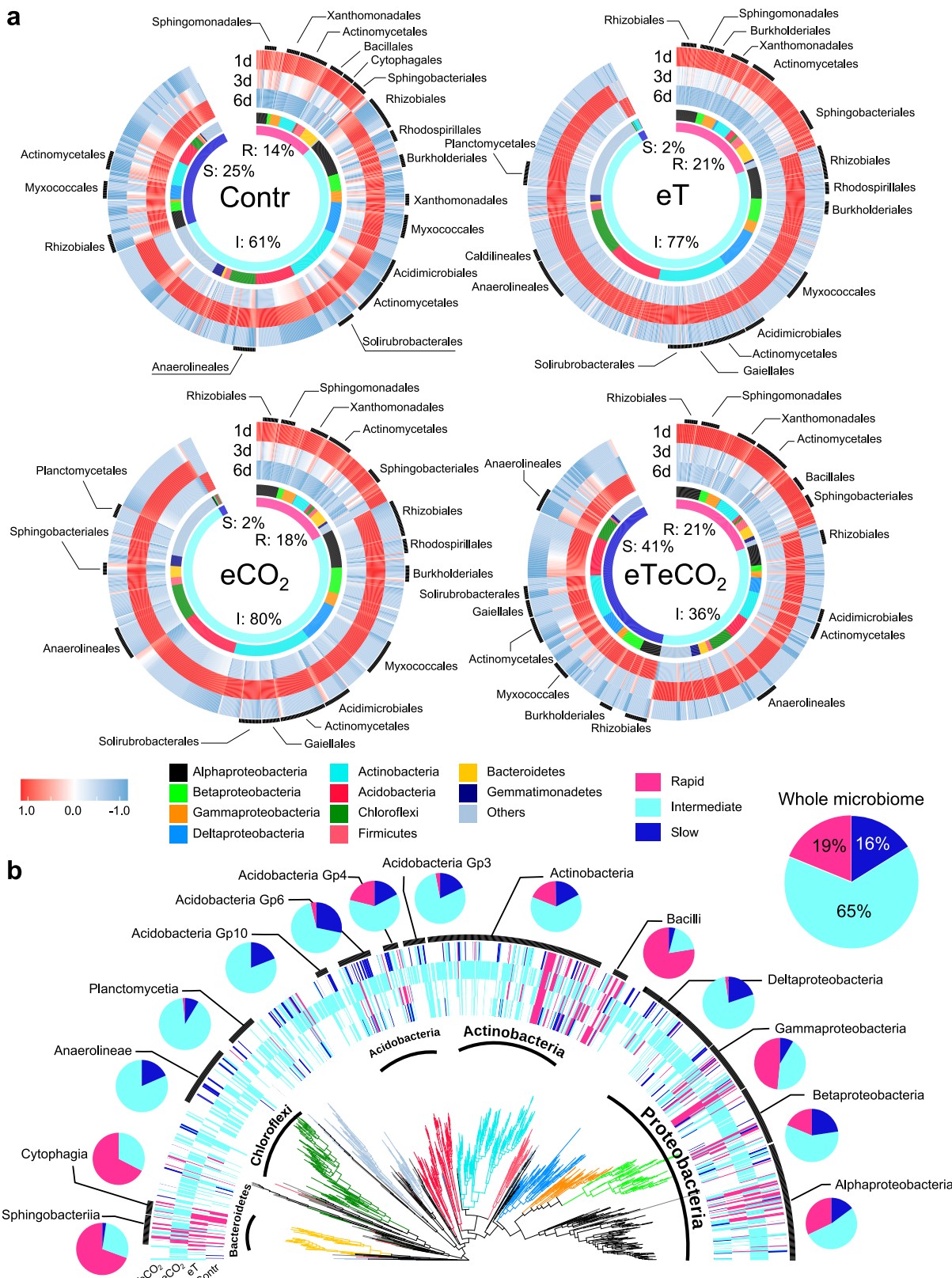

**Fig. 3 | Taxa with distinct growth strategies.** We defined three growth strategies: rapid, intermediate, and slow responses. The heatmap showed the growth dynamics of taxa across the incubation period (the values were Z-score standardized and ranged from −1 to 1) (**a**). Growth strategies of bacterial taxa in relation to phylogeny (**b**). The pie charts represent the proportion of responders at the class level using data from all four treatments. The total proportion of growth response strategies in this incubation experiment is also shown. R: rapid response; I: intermediate response; S: slow response. Contr represents the ambient temperature and $CO_2$ concentration; eT represents warming only; $eCO_2$ represents elevated $CO_2$ concentration only; and $eTeCO_2$ represents combined warming and $CO_2$ enrichment.

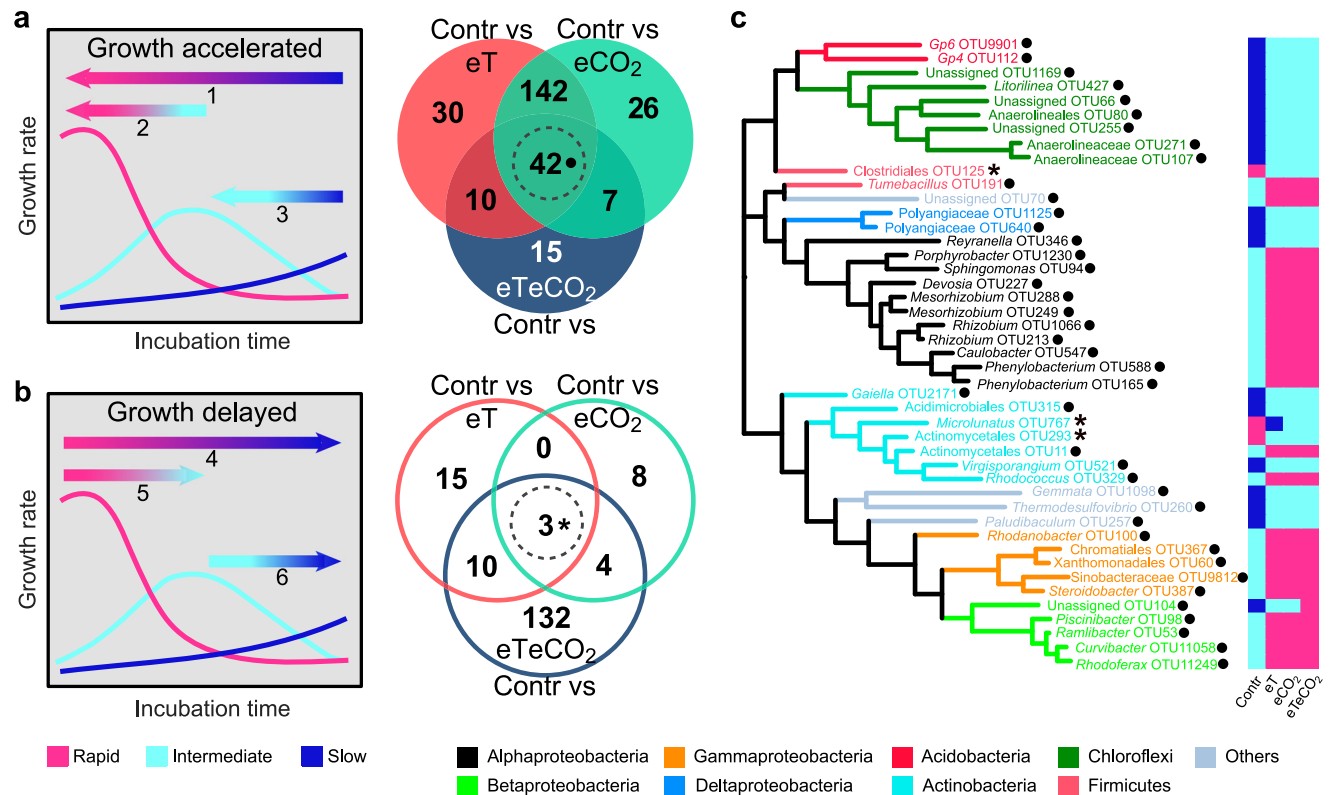

**Fig. 4 | Climate change shifted growth strategies of soil bacteria.** Microbial growth accelerated (**a**) or delayed (**b**) by climate change. The 'growth accelerated' included three situations, (1) microbial growth strategies changed from slow to rapid, (2) from intermediate to rapid, and (3) from slow to intermediate when comparing ambient conditions (Contr) with the other three climate treatments; The 'growth delayed' shift included the changed strategies (4) from rapid to slow, (5) from rapid to intermediate, and (6) from intermediate to slow. Veen plots showed the overlaps of taxa with changed strategies among the comparisons of the three climate change scenarios with Contr. Closed circles: growth accelerated;

Open circles: growth delayed. Subsequently, 45 core strategy-changed taxa (42 OTUs were always 'growth accelerated' and 3 OTUs were always 'growth delayed' in the three paired comparisons, circle with dashed lines) were selected for phylogenetic analysis (**c**). Black dot following OTU labels: this OTU belonged to 'growth accelerated'; Star: this OTU belonged to 'growth delayed'. Contr represents the ambient temperature and $CO_2$ concentration; eT represents warming only; $eCO_2$ represents elevated $CO_2$ concentration only; and $eTeCO_2$ represents combined warming and $CO_2$ enrichment.

(Fig. 4c), the 42 consistently "growth accelerated" OTUs showed a phylogenetically widespread distribution: across eight dominant phyla. Most OTUs affiliated with Alphaproteobacteria, Betaproteobacteria, and Gammaproteobacteria changed their strategies from "intermediate" to "rapid". The OTUs affiliated with Deltaproteobacteria, Actinobacteria, Chloroflexi, and Acidobacteria mainly changed the strategies from "slow" to "intermediate" responses after acclimation to climate change.

To explore the impacts of climate change on the phylogenetic patterns of growth strategies, the nearest taxon index (NTI) was used. All categories, with the exception of intermediate responders in eT, were clustered at phylogenetic branches (NTI > 0, $p < 0.05$, Table 2). The patterns were modified by climate change. For example, rapid responders in three climate change scenarios (i.e., eT, $eCO_2$, and $eTeCO_2$) had consistently lower NTI than that in the control soil, indicating weakened phylogenetic conservation. The phylogenetic pattern of intermediate responders in the control soil was clustered, but became random under climate warming. Furthermore, the three additional phylogenetic indices—phylogenetic dispersion, Blomberg's K and Pagel's λ—were similar to the result of the NTI index, showing phylogenetic signals in most growth strategies ($p < 0.05$). Phylogenetic signals of rapid responders were weakened after climate change (Supplementary Table 1). Altogether, acclimation to climate change reshaped microbial growth patterns by shifts in species composition and in their response strategy, which ultimately influenced the phylogenetic patterns.

## Phylogenetic constraints and environmental acclimation explain the growth response patterns

We applied variance partitioning analysis to quantify the contributions of phylogenetic constraints and environmental acclimation to three growth strategies. We used an abstract representation of phylogeny and environment as principal components (PCs), considering the dimensionality, collinearity, and redundancy of this data. The respective relationships were approximated by a linear model, which helped to assess the variance of growth strategies explained by phylogenetic constraints and environmental acclimation (see "Methods" for details). These constraints and acclimation affected the three growth strategies differently (Supplementary Fig. 6): The strongest phylogenetic effects were observed for rapid responders (17%), followed by intermediate responders (7%), whereas phylogeny had a little effect on slow responders (0.3%). There was an opposite trend in the effects of environmental acclimation on growth strategies: Environment had a substantial impact on the distribution of slow responders (72%) across taxonomic groups and a slight effect on rapid responders (3%).

## Discussion

Phylogenetic relatedness has a substantial impact on trait-based patterns of microbes[11,12], yet the degree to which their eco-physiological traits are affected by phylogeny is unknown. Microorganisms obtain genomes from their parent cells, reflecting their phylogeny. The growth-related genes mainly belong to the core genome, which is

**Table 2 | The nearest taxon index (NTI) of three growth strategies across climate change treatments**

| Field treatment | Strategy | Num taxa | NTI | p-value |
|---|---|---|---|---|
| Contr | Rapid | 107 | 7.3 | **0.001** |
| | Intermediate | 453 | 2.08 | **0.021** |
| | Slow | 190 | 2.43 | **0.007** |
| eT | Rapid | 184 | 5.79 | **0.001** |
| | Intermediate | 675 | −0.23 | 0.583 |
| | Slow | 14 | 2.52 | **0.009** |
| eCO2 | Rapid | 159 | 4.29 | **0.001** |
| | Intermediate | 684 | 3.53 | **0.001** |
| | Slow | 13 | 1.73 | **0.046** |
| eTeCO2 | Rapid | 142 | 5.42 | **0.001** |
| | Intermediate | 241 | 2.36 | **0.007** |
| | Slow | 287 | 5.24 | **0.001** |

A permutation test was used for data analysis and the p-value based on comparison of phylogenetic distance observed and that based on 1000 permutations of a null model. Values in bold: $p < 0.05$. Low p-values (and positive indices) indicate that species are more phylogenetically related than expected by chance (clustered). NTI, nearest taxon index. Contr represents the ambient temperature and $CO_2$ concentration; eT represents warming only; $eCO_2$ represents elevated $CO_2$ concentration only; and $eTeCO_2$ represents combined warming and $CO_2$ enrichment.

mostly inherited vertically and reflects the phylogeny[31]. This explains the phylogenetic signals of the three growth strategies (Table 1), which supports the first hypothesis that phylogeny influences the growth strategies of bacterial species after soil wet-up. The effects of environmental selection on microbial strategy, however, cannot be ignored[16,32]. Phenotypic plasticity is crucial in the physiological performance of microorganisms to acclimate to broad and varied environments[33]. Furthermore, microorganisms can also obtain adaptive genes from other species through horizontal gene transfer[32]. These factors could drive these differences of microbial ecophysiological traits.

The phylogenetic conservation of growth strategies is also evident on the phylogenetic tree (Fig. 3b). For example, most members of bacterial classes, e.g., 78% of Bacilli OTUs, 69% of Sphingobacteriia OTUs, and 67% of Cytophagia OTUs were classified as rapid responders. Nonetheless, taxon-specific traits are not uniformly shared by all taxon members due to genome divergence[5,6,8]. Moreover, the selection caused by environmental factors can also confound the ecological coherence, resulting in various functional phenotypes even for bacteria that share a common evolutionary history[11].

Elevated temperature and $CO_2$ shifted the strategy within the species level: The growth of many taxa was accelerated under warming or $CO_2$ enrichment solely, while the growth response was mostly delayed by a simultaneous increase of temperature and $CO_2$ (Fig. 4a, b). These results confirm our second hypothesis that warming and $CO_2$ enrichment shift the pre-existing growth strategies of certain species. Furthermore, we detected negative density-dependent selection under warming or elevated $CO_2$ (both $p < 0.01$, Supplementary Fig. 5), which is common under resource limitation[28]. A general concept is that this growth is proportional to the rate of microbial resource gain[34], and previous work has shown that under resource limitation, per capita resource allocations declined with increased population densities, which decreased the growth rates[28]. Interestingly, the strength of the density-dependence became very weak and non-significant under simultaneous elevated $CO_2$ and warming. Note that the $R^2$ values between species growth rates and densities in eT and $eCO_2$ treatments were low (0.037 and 0.062, respectively), pointing to other driving forces shaping microbial growth.

Warming affects C storage and nutrient cycling in soils by influencing plant and microbial processes[2]. Microbes are highly sensitive to

temperature, and warming generally raises microbial activity and respiration, which decreases the quantity and quality of organic matter in soil[7,35–37]. Earlier results of this long-term experiment showed that warming diminished aboveground plant biomass and grain yield, indicating decreased plant-derived C input and N demand[25,26]. Conversely, soil nitrate content increased under warming because of faster SOM and plant residue mineralization, leading to $CO_2$ release and mineral N accumulation[2]. Considering these results, we speculate that a reduced soil C-to-N ratio could increase C limitation for microorganisms after long-term warming.

Contrary to the negative effect of warming in our FACE system, an elevated $CO_2$ concentration increased plant biomass[25]. This in turn increased the belowground input of photosynthates and the N demand of plants[25,38]. Increased labile C stimulated bacterial growth[39], which further intensified the competition for N between plants and microorganisms[40,41]. Indeed, elevated $CO_2$ concentrations substantially reduced the soil nitrate content and increased N limitation for microorganisms[2,42]. Consequently, the resource limitations caused by warming and $CO_2$ enrichment were opposite: stronger C limitation by warming but increased N limitation by $CO_2$ enrichment. Stoichiometric imbalance or stress limit microbial growth, requiring more investments to maintain essential metabolic functions, or leading to dormancy[43,44]. Species with higher population densities may be more strongly limited under resource-scarce conditions compared to species with lower population densities[28]. This partly explains the negative density dependence in eT and $eCO_2$ scenarios. As these opposite effects are likely to offset each other, the strength of density dependence was weakened under the simultaneous warming and $CO_2$ enrichment scenario.

The ability to grow rapidly is crucial in competing for limited resources, especially under stoichiometric imbalance[8,14]. Accordingly, taxa with accelerated growth response are environmentally selected by warming-only or $CO_2$ enrichment-only. Conversely, mitigating resource limitations reduced the role of the selection (Supplementary Fig. 7)[28,45,46], and slow responders were more abundant under combined manipulations of temperature and $CO_2$ (Fig. 3a). This underlines that the interactions between warming and $eCO_2$ are very complex because their individual effects are partly opposite. This urgently calls for more long-term multifactor manipulation experiments to dissect the variable responses of multiple ecological functions and to accurately predict the effects of future global change.

Phylogeny and environment jointly influence microbial phenotypes[23,31]. We identified how these two factors shape microbial growth dynamics (Supplementary Fig. 6). Phylogenetic constraints and environmental acclimation explained the growth strategies, whereby rapid responders were chiefly governed by the former, and slow responders by the latter. Rapid microbial growth is based on combinations of many physiological traits including (i) cellular self-triggering (for immediate resuscitation), (ii) DNA replication and protein synthesis, and (iii) resistance to osmotic pressure[8,47]. These essential growth-related genes tend to belong to the core genomes[48], which usually evolved slowly and were unlikely to be transferred among microorganisms[49]. This means that the microbial trait of rapid growth was likely inherited vertically from ancestors to descendants. Unexpectedly, less than 20% of variation is explained by these variables for the rapid and intermediate responders, which could reflect other factors shaping the growth strategies. For instance, historical contingencies may affect microbial community composition[50], which further affected the current community functions as well as species' trait expressions. The metabolic processes could be influenced by the co-limitation of multiple resources, which complicates the prediction of microbial physiological performance[51,52]. Lastly, the undetected changes of soil micro-environment and stochastic processes (e.g., genetic drift), may equally contribute to the observed unexplained components.

Most microbial traits are based on some degree of phylogenetic conservation and, consequently, the significance of phylogeny and their respective functional traits should be considered when studying microbial-environment relationships[12]. The three strategies of microbial growth response to wet-up events are mostly phylogenetically dependent, and the strength of phylogenetic signals depends on the growth response types. Such a differentiation is probably due to distinct contributions of phylogeny and environmental acclimation to the eco-physiological traits of soil microorganisms under given climate conditions. Warming and $eCO_2$ shift the growth dynamics within a given species locally and, potentially, globally. Such a shift in microbial eco-physiology could further affect biogeochemical cycling of elements in soil and climate feedback. Collectively, even though the growth strategies are conserved phylogenetically, the acclimation to a new environment shifts the strategies, and this mainly for the slow responders. Our results add new evidence that phylogeny and environmental acclimation are essential to understand the evolution of growth traits of soil microorganisms and their response to climate change.

## Methods

### Study site, field experiment, and soil sampling
This study relied on an experimental field station that simulates atmospheric $CO_2$ enrichment and warming (Fig. 1a). The experimental site is located at Kangbo village (31°30′N, 120°33′E), Guli Township, Changshu municipality in Jiangsu Province, China. The climate type is a subtropical monsoon climate with a mean annual precipitation between 1100–1200 mm and an annual average temperature of approximately 16 °C. The high rainfall and temperature mainly occur from May through September[25]. The soils are Gleyic Stagnic Anthrosols derived from clayey lacustrine deposits. The main properties of the topsoil (0–15 cm) before the field experiment were as follows: pH ($H_2O$) 7.0, bulk density 1.2 g $cm^{-3}$, and contents of organic C and total N of 16.0 g $kg^{-1}$ and 1.9 g $kg^{-1}$, respectively.

The FACE system has been well described in previous publications[26,53]. The climate change treatments were set according to the representative concentration pathway (RCP) scenario that modeled $CO_2$ atmospheric concentration and temperature elevation in approximately 30–40 years. Elevated atmospheric $CO_2$ concentration and warming of crop canopy air were maintained 24 h a day during the crop-growing period. Each treatment was replicated in three rings with the same infrastructure, and the rings were arranged in a split row design (Fig. 1b). All rings were buffered by adjacent open fields to avoid any treatment cross-over. Rice and wheat cultivations of all plots were managed with local conventional practices. Soil samples for qSIP incubation were collected in June 2020.

### Evaluation of nutrient pulses in soil after water addition
To determine if there is a nutrient pulse after soil rewetting, we estimated the dynamics of several biochemical characteristics (i.e., $CO_2$ fluxes, hydrolase activity, DOC, DN, TC, and TN) in soils before and during the incubation. Soil samples for nutrient flux measurements were collected from the free-air $CO_2$ enrichment and warming experimental station in July 2022.

To measure the $CO_2$ production rate, soil (2.00 g) was set in a 29 mm diameter glass vial (Volume 23 mL). Water (400 μL) was added evenly to soil in each vial. Gas samples were then collected from each incubation vial at multiple time points (i.e., 3 h, 6 h, 9 h, 12 h, 24 h, 72 h, and 144 h after water addition), and $CO_2$ concentrations were determined in all gas samples with a Trace Gas Chromatograph (Agilent 7890, Santa Clara, CA, USA). Other biochemical characteristics were measured in parallel incubations, with destructive sampling. Briefly, soil (40.00 g) was set in a plastic jar, and 8 mL water was added evenly to the soil. All the incubation vials and jars were placed in the dark at 25 °C and soil moisture content was adjusted twice a day. Destructive

sampling was conducted after 1, 3, and 6 days of incubation. Including prewet soil (the soil before water addition), a total of 48 soil samples were obtained. Soil samples were weighed before and after being oven dried at 105 °C, and soil moisture content was measured using the gravimetric method. Fluorescein Diacetate (FDA) hydrolase activity of soil was measured by using soil FDA hydrolase activity assay kit (Solarbio®, Beijing, China) according to the manufacturer's protocol. Soil total carbon and total nitrogen were measured with a C/N elemental analyzer (multi EA® 5000, analytikjena, Germany). Dissolved organic carbon and dissolved nitrogen were analyzed using a TOC/TN analyzer (multi N/C® 3100, analytikjena, Germany) after extraction with distilled water.

### Simulating nutrient pulse after rewetting dry soil by adding $^{18}$O-water
To determine whether and how microbial growth varied in response to pulse events after long-term acclimation to warming and $CO_2$ enrichment, we estimated the population-specific growth rates of active microbes by conducting a $^{18}$O-water incubation experiment combined with DNA quantitative stable isotope probing (DNA-qSIP) (Fig. 1c). The incubation conditions were similar to those reported in a previous study[6]. In brief, approximately 60 g fresh soil of each treatment were sieved (2 mm) and air-dried (24 h at room temperature) immediately after transport to the laboratory. Then, triplicate samples of dry soils (2.00 g) were incubated in the dark at room temperature in sterile plastic aerobic culture tubes (17 × 100 mm) with 400 μL of 98 atom% $H_2^{18}O$ or natural abundance water ($H_2^{16}O$) for 6 days, with harvests at four time points (T = 0, 1, 3, 6 d after water addition). DNA was extracted from the soils of 0 d incubation treatment immediately after water addition (-30 s interval), representing the prewet treatment. At each harvest, soils were destructively sampled and immediately stored at −80 °C. A total of 84 soil samples (4 climate change treatments × 3 replicates at 0 h with $H_2^{16}O$ addition + 4 treatments × 3 subsequent time points × 3 replicates × 2 types of $H_2O$ addition) were collected.

### DNA extraction and isopycnic centrifugation
Total DNA from all the collected soil samples was extracted using the FastDNA™ SPIN Kit for Soil (MP Biomedicals, Cleveland, OH, USA) according to the manufacturer's instructions. The concentration of extracted DNA was determined fluorometrically using Qubit® DNA HS (High Sensitivity) Assay Kits (Yeasen Biotechnology, Shanghai, China) on a Qubit® 4 fluorometer (Thermo Scientific™, Waltham, MA, USA). The DNA samples of day 1, day 3, and day 6 were used for isopycnic centrifugation, and the detailed pipeline was described previously with minor modifications[6]. Briefly, 3 μg DNA were added into 1.85 g $mL^{-1}$ CsCl gradient buffer (0.1 M Tris-HCl, 0.1 M KCl, 1 mM EDTA, pH = 8.0) with a final buoyant density of 1.718 g $mL^{-1}$. Approximately 5.1 mL of the solution was transferred to an ultracentrifuge tube (Beckman Coulter QuickSeal, 13 mm × 51 mm) and heat-sealed. All tubes were spun in an Optima XPN-100 ultracentrifuge (Beckman Coulter) using a VTi 65.2 rotor at 177,000 × $g$ at 18 °C for 72 h with minimum acceleration and braking.

Immediately after centrifugation, the contents of each ultracentrifuge tube were separated into 20 fractions (-250 μL each fraction) by displacing the gradient medium with sterile water at the top of the tube using a syringe pump (Longer Pump, LSP01-2A, China). The buoyant density of each fraction was measured using a digital hand-held refractometer (Reichert, Inc., Buffalo, NY, USA) from 10 μL volumes. Fractionated DNA was precipitated from CsCl by adding 500 μL 30% polyethylene glycol (PEG) 6000 and 1.6 M NaCl solution, incubated at 37 °C for 1 h, and then washed twice with 70% ethanol. The DNA of each fraction was then dissolved in 30 μL of Tris-EDTA buffer. Detailed information (company names and catalog numbers) of the reagents and consumables for qSIP experiment is provided in Supplementary Data 2.

## Quantitative PCR and sequencing

Total 16S rRNA gene copies for DNA samples of all the fractions and the day-0 soils were quantified using the primers for V4-V5 regions: 515F (5′-GTG CCA GCM GCC GCG G-3′) and 907R (5′-CCG TCA ATT CMT TTR AGT TT-3′)[54]. Plasmid standards were prepared by inserting a copy of purified PCR product from soil DNA into *Escherichia coli*. The *E. coli* was then cultured, followed by plasmid extraction and purification. The concentration of plasmid was measured using Qubit DNA HS Assay Kits. Standard curves were generated using 10-fold serial dilutions of the plasmid. Each reaction was performed in a 25-µL volume containing 12.5 µL SYBR Premix Ex Taq (TaKaRa Biotechnology, Otsu, Shiga, Japan), 0.5 µL of forward and reverse primers (10 µM), 0.5 µL of ROX Reference Dye II (50×), 1 µL of template DNA, and 10 µL of sterile water. A two-step thermocycling procedure was performed, which consisted of 30 s at 95 °C, followed by 40 cycles of 5 s at 95 °C, 34 s at 60 °C (at which time the fluorescence signal was collected). Following qPCR cycling, melting curves were obtained to ensure that the results were representative of the target gene. Average PCR efficiency was 96% and the average slope was −3.38, with all standard curves having $R^2 \geq 0.99$.

The DNA of day-0 samples (unfractionated) and the fractionated DNA of fractions with buoyant density between 1.695 and 1.735 g/mL were selected for 16S rRNA amplicon sequencing by using the same primers of qPCR (i.e., 515F/907R). Eleven out of 20 fractions from each ultracentrifuge tube (density between 1.695 and 1.735 g/mL) were selected because they contained more than 99% gene copy numbers of the 20 fractions. A total of 804 DNA samples (12 unfractionated DNA samples + 72 × 11 fractionated DNA samples) were sequenced using the NovaSeq6000 platform (Genesky Biotechnologies, Shanghai, China).

The sequences were quality-filtered using the USEARCH v.11.0[55]. In brief, sequences < 370 bp and total expected errors > 0.5 were removed. Chimeras were identified and removed. Subsequently, high-quality sequences were clustered into operational taxonomic units (OTUs) using the UPARSE algorithm at a 97% identity threshold, and the most abundant sequence from each OTU was selected as a representative sequence. The taxonomic affiliation of the representative sequence was determined using the RDP classifier (version 16)[56]. In total, 51,127,459 reads of the bacterial 16S rRNA gene and 11,898 OTUs were obtained. The 16S rRNA amplicon sequences were uploaded to the National Genomics Data Center (NGDC) Genome Sequence Archive (GSA) with accession number CRA006507.

## Quantitative stable isotope probing calculations

We used the amount of [18]O incorporated into DNA to estimate the growth rates of active taxa[24,57]. The density shifts of OTUs between [16]O and [18]O treatments were calculated following the qSIP procedures[24]. Briefly, the number of 16S rRNA gene copies per OTU in each density fraction was calculated by multiplying the OTU's relative abundance (acquisition by sequencing) by the total number of 16S rRNA gene copies (acquisition by qPCR). Then, the GC content and molecular weight of a particular taxon were calculated. Further, the change in [18]O isotopic composition of 16S rRNA genes for each taxon was estimated. We assumed an exponential growth model over the course of the incubations, and absolute population growth rates were estimated over each of the three time intervals of the incubation: 0–1, 0–3, and 0–6 d corresponding to the samplings at days 1, 3, and 6. The absolute growth rate is a function of the rate of appearance of [18]O-labeled 16S rRNA genes. Therefore, the growth rate ($g$) of taxon $i$ was calculated as:

$$g_i = \ln\left(\frac{N_{\text{TOTAL}it}}{N_{\text{LIGHT}it}}\right) \times \frac{1}{t} \tag{1}$$

Where $N_{\text{TOTAL}it}$ is the number of total gene copies for taxon $i$ and $N_{\text{LIGHT}it}$ represents the unlabeled 16S rRNA gene abundances of taxon $i$ at the end of the incubation period (time $t$). $N_{\text{LIGHT}it}$ is calculated by a function with four variables: $N_{\text{TOTAL}it}$, molecular weights of DNA (taxon

$i$) in the [16]O treatment ($M_{\text{LIGHT}i}$) and in the [18]O treatment ($M_{\text{LAB}i}$), and the maximum molecular weight of DNA that could result from assimilation of $H_2^{18}O$ ($M_{\text{HEAVY}i}$)[24]. We further calculated the average growth rates (represented by the production of new 16S rRNA gene copies of each taxon per g dry soil per day) over each of the three time intervals of the incubation: 0–1, 0–3, and 0–6 d, using the following equation[39]:

$$\frac{dN_i}{dt} = N_{\text{TOTAL}it}\left(1 - e^{-g_i t}\right) \times \frac{1}{t} \tag{2}$$

Where $t$ is the incubation time (d). All data calculations were performed using the qSIP package (https://github.com/bramstone/qsip) in R (v. 3.6.2).

## Grouping of taxa into growth strategies

We compared the average growth rates of taxa at three time intervals ($n = 3$ in each time interval) and classified the species into rapid, intermediate, and slow growth strategies based on the timing of the maximum growth rate (Fig. 1d): (1) Rapid responders: Species had the highest growth rates by 1 day of the incubation; (2) Intermediate responders: Species had the highest growth rates at the 3-day incubation; (3) Slow responders: Species had the highest growth rates at the 6-day incubation. The taxa with growth rates significantly greater than zero can be divided into one of three strategies in each treatment.

## Analyses of phylogenetic conservation

Phylogenetic tree analyses were performed in Galaxy/DengLab (http://mem.rcees.ac.cn:8080/) with PyNAST Alignment and FastTree functions[58,59]. The trees were visualized and edited using iTOL[60]. To estimate the phylogenetic patterns of growth strategies, phylogenetic dispersion (D) was calculated by the function 'phylo.d' of package "caper" in R (v. 3.6.2). The D statistic equal to 1 means the observed trait has a phylogenetically random distribution across the tips of the phylogeny, and 0 means the observed trait is as clumped as if it had evolved by Brownian motion[29]. Increasing phylogenetic clumping in the binary trait is indicated by values of D decreasing from 1. The $p$ values were obtained by performing 1000 permutations to test D for significant departure from 1 (random distribution). Furthermore, the phylogenetic signal metrics Blomberg's K and Pagel's λ, were used to test for significantly nonrandom phylogenetic distributions using the package "phylosignal" in R. The $p$ values of both indices were used to test the significance of phylogenetic signals.

The nearest taxon index (NTI) was calculated to determine the degree of phylogenetic clustering as described previously[5]. The mean nearest taxon distance (MNTD) were calculated in the "picante" package of R[61]. The values of NTI are equivalent to the negative output of the standardized effect size (SES) of the observed MNTD distances, which test whether the distribution of growth strategies across different phylogenetic groups is random or nonrandom. NTI values > 0 and their $p$ values < 0.05 represent phylogenetic clustering, while NTI values < 0 and $p$ values > 0.95 indicate phylogenetic over-dispersion. The $p$ values of NTI between 0.05 and 0.95 represent random phylogenetic distributions. The data were converted into binary matrices (1 s and 0 s) before all phylogenetic analyses. For the OTUs that were present in more than one treatment and had differing responses (in the analyses when all climate treatments are combined, i.e., in Table 1 and Supplementary Fig. 6), the OTU was classified to the respective growth responder when that taxa exhibited that specific growth strategy in any treatment.

## Quantification of explained variance in the distribution patterns of strategies

The phylogenetic distance matrix obtained from the phylogenetic tree and three binary matrices (including four climate treatments) for three growth strategies were decomposed by the package 'FactoMineR' in R.

The distribution matrices (binary) of three growth responders in the four climate treatments were used to represent the impact of $CO_2$ and temperature on bacterial growth. To predict the microbial growth strategy (i.e., "y", a 1 or 0 indicating the membership in a response category), variance partitioning was performed by using the first four phylogenetic principal components (PCs) (accounting for over 80% of phylogenetic variance) and two environment PCs (accounting for over 65% of habitat variance). The fraction of the variance explained by phylogenetic constraints and environmental acclimation was calculated by the package 'car' in R.

## Statistical analyses

Uncertainty of growth rates (95% confidence interval) was estimated using a bootstrapping procedure with 1000 iterations[6]. The cumulative growth rates at phylum-level were estimated as the sum of taxon-specific growth rates of those OTUs affiliated to the same phylum. The per capita growth rates of each OTU were calculated by dividing absolute growth rates by the total 16S rRNA gene abundance of taxon $i$. Ecological processes of active populations (e.g., density-dependence) after rewetting dry soil were estimated by correlating species-specific fitness (refers to per capita growth rates of each OTU) with initial population size using linear regression analyses (R v.3.6.2, 'lm' function). The beta diversity of the bacterial community at 0 d incubation was visualized by unconstrained principal coordinate analysis (PCoA) and tested by permutational multivariate analysis of variance (PERMANOVA) based on Bray-Curtis distance, using the vegan package in R (v. 3.6.2)[62]. Comparisons between climate scenarios were tested by two-way ANOVA and LSD post hoc tests (SPSS 19 for Windows, IBM Corp., Armonk, NY, USA).

## Reporting summary

Further information on research design is available in the Nature Portfolio Reporting Summary linked to this article.

# Data availability

The sequence data generated in this study have been deposited in the National Genomics Data Center (NGDC) Genome Sequence Archive (GSA) under accession code CRA006507.

# Code availability

R code for qSIP pipeline and statistical analyses can be accessed from https://zenodo.org/badge/latestdoi/484023335.

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

## Acknowledgements
This work was supported by the National Key R&D Program of China [2021YFD1900300 (N.L.)] and the National Natural Science Foundation of China [42277100 (N.L.), U2003210 (S.W.G.)].

## Author contributions
N.L. contributed to the conceptualization and supervision of the experiment and edited the manuscript; Y.R. conducted the experiment and data analysis, and wrote the original draft; X.Y.L. and X.H.Z. provided samples; Q.R.S. and S.W.G. contributed to project scope; Y.K., N.L., X.Y.L., Y.F.Y., J.J.G., Q.C.X., and Y.R. contributed the review and editing of manuscript.

## Competing interests
The authors declare no competing interests.
