## [Peer Review File · Nature Communications]

Elevated temperature and CO₂ strongly affect the growth strategies of soil bacteriaREVIEWER COMMENTS

Reviewer #1 (Remarks to the Author):

The authors determined microbial population-specific growth rates by ¹⁸O-DNA quantitative stable isotope probing (¹⁸O-qSIP) in soils subjected to a free-air CO₂ enrichment (FACE) combined with warming. The soils were sampled from the FACE experiment in Kangbo village Guli Township, Changshu municipality in Jiangsu Province, China, and the experiment started in 2010. The authors found that bacterial taxa could be separated into three distinct groups with different growth strategies which were phylogenetically conserved in soils that were not manipulated with higher CO₂ or temperature environments but when these manipulations were implemented the phylogenetic signal disappeared.

The work appears to be well and carefully done and the data and analyses are of great value to the larger microbial ecological community. There are very few studies in which the growth rates of bacterial populations from FACE experiments have been investigated via qSIP and the large quantity of data that comprises the study described in this manuscript, will substantially advance our knowledge of bacterial responses to elevated CO₂ and temperature manipulations. However, I think the manuscript will require major changes and a considerable rewrite.

My chief complaint about the manuscript is that it is too long because of redundant information. For instance, L114 describes the soils that were collected in the results section, it is again described in L432 and in the legends of various figures. Similarly, the discussion section has many sentences that restate findings presented in the Results section. Another example is the description of growth strategies (L192 to 204) which occurs repeatedly in the manuscript. It is important to write a concise manuscript in which information is only presented once.

I also find that the discussion section is too long and while it contains presentation of other studies, these findings are often not well related to the growth rates of bacteria measured by the authors. For instance, the authors did not measure the number of rRNA genes on a taxa's genome and so there appears limited need to extensively discuss relationships between rRNA copy number and growth rates. If there are difficulties relating other studies to the authors findings, the discussion of those studies should be removed from the paper.

I think the introduction focuses too much on the "Birch effect". While soil wetup effects are well documented, they always occur when liquids, be it water or a glucose solution, are added to soil. I think in soils that routinely dry and wet back up, unlike soils from Mediterranean climates such as California, have minimal Birch effects upon rewetting. The paper is not really about the "Birch effect", it is about the impact of CO₂ and temperature on bacterial growth rates in soil and the introduction should focus on that.

I suggest changing the title of the manuscript from "Phylogenetic constraint and environmental acclimation link to soil microbial growth strategies under CO₂ enrichment and warming." To something like: "Elevated temperature and CO₂ strongly affect growth strategies of soil bacteria."

More minor edit suggestions:

Abstract: I suggest deleting the first few sentences of the abstract and start with the sentence in Line 25.

L135 insert "the" before "6"

L43 Delete "The evidence is overwhelming that"

L159 change "the" to "a" before phylogenetic

L217 change "shared with three comparisons" to "shared among the three treatments"

I struggle with the following argument: "The essential genes for microbial growth are considered to be

the core genome, which is generally inherited vertically and reflects phylogeny strictly. This helps explain the observed phylogenetic conservation of the strategies related to growth response.” How do the authors know that the core genome allows microorganisms to grow faster or slower (i.e. be more competitive)?

L280 Suggest change of “visually evident based” to “evident”.

L293 Suggest changing “validated” to “supported”

L295 Suggest deleting “This helps grasp the relationship between phenotypic characteristics and the phylogeny of microorganisms.

L 309 Does the sentence “Microbes possessed metabolic plasticity for carbon use to suit resource availability or interspecies interactions, which promoted or reduced microbial growth^{28, 29, 30} refer to microorganisms in this study? If not, that should be clearly indicated.

L310 “extremely high intraspecific variation of the genome” It is unclear what genome this sentence refers to

L355 suggest changing “for resources under limitation” to “for limited resources”.

L357 suggest changing “mitigated” to “mitigating”

L370 suggest changing “microorganismic” to “microbial”.

L442 “The incubations were described in detail previously⁶.” I think this refers to another study and not the actual incubations in this study.

L513 suggest deleting “As ¹⁸O labeling occurs during cell growth via DNA replication”

Figures:

I am unclear what “low growth” means in bacterial phyla list of Figure 2

Figure 5 The results presented in Figure 5 are described in the text and do not require an entire figure.

Figure 6 seems to recap information from other figures and could also be removed from the manuscript

Reviewer #2 (Remarks to the Author):

The manuscript by Ruan et al. describes the growth responses of soil bacteria following rewetting in soils from a long-term global change experiment. The authors used qSIP with ¹⁸O water to measure microbial growth rates after 1, 3, and 6 days. Based on the day of maximum growth taxa were classified as “rapid”, “intermediate” or “slow” responders. In general, the experimental design and methods appear to be of sufficient quality. The results are likely to be of interest to the microbial ecology community and may help advance our understand of how microbial systems will respond to global change.

I do have some significant concerns about some of the data analyses, the framing of the paper, and the interpretation of the results.

1) Why are the treatments grouped together for the phylogenetic dispersion, Blomberg’s K, and Pagel’s lambda analysis but not the NTI analysis? For the analyses in table 1 how did you treat OTUs that were present in more than one treatment and had differing responses? Were the data converted into binary (1s and 0s) before all the phylogenetic analyses? These details should be explained in the manuscript.

2) The authors frequently conflate the categorized growth responses they observed with ecological or life strategies. This study focuses on one trait the timing of maximum growth after rewetting. One trait cannot be used to assign an ecological strategy as ecological strategies are defined by a suite of traits that covary due to eco-evolutionary tradeoffs.

3) The authors seem to confuse adaptation, acclimation, and plasticity in their interpretation of changes in the timing of maximum growth following rewetting. As with “ecological strategy” these terms are too often used and abused in the field of microbial ecology perhaps contributing to their misuse in this paper. They suggest that communities and OTUs have adapted to the global change factors via changes in their genome but provide no evidence for this. It’s far more likely that the global change factors altered the soil environment by changing plant and microbial processes that impact water, carbon, and nitrogen availability) which in turn impact the timing of microbial growth following soil rewetting. Changes in the trait (timing of growth) in response to global change factors indicate phenotypic plasticity that enables the organisms to acclimate to the altered environment.

Unfortunately, the authors provide little insight into how changes in the soil could be mediating the impacts of eCO₂ and temperature on the timing of microbial growth. At a minimum the environmental data used for the analysis presented in Figure 5 should be included in the manuscript.

4) The analysis in Fig 5, what exactly is being predicted? What is “y” ? Is it a 1 or 0 indicating the membership in a response category? Or is it the growth rate if the OTU is in the response category?

Specific comments

Introduction – The introduction does not adequately establish why understanding the influence of phylogeny on function is important.

Line 126 – I suggest you change “adapted” to “acclimated”

Line 42 – Remove “conserved”

Line 144 – Fig 2 shows proportions but is labeled as (%). The percentages are a little confusing. I think it would help if you reminded the reader how to interpret them. For instance, 29% of new 16S rRNA gene copies.

Line 172 This is incorrect – Blomberg’s K can be greater than 1. Make sure you fully understand the statistics you are employing.

Line 176 If a trait is conserved it should have a higher phylogenetic signal than expected under Brownian motion ($k > 1$). This is not the case for the responses observed, there is an influence of phylogeny but that does not indicate it’s conserved.

Line 226-236 The Picrust analysis and it’s results here and the heat map in Fig 4c are not well integrated with the rest of the manuscript. They feel tacked on and do not meaningfully enhance the manuscript. I suggest they be removed as they detract from the focus of the paper.

Line 237 This section would fit better with the other phylogenetic analysis.

Line 294 “unifying life strategies” , this language is far too strong to described the responses you measured.

Line 307 You have no evidence that environmental section is acting on the trait you measured.

Lines 311 – 315 I think this interpretation is too strong. There was not statistical analysis to show that rates were significantly higher or lower across the 3 time points thus while the categorization may have changed it’s unclear whether this is a truly meaningful change in the timing of growth. Even if the time of growth did change, this is not necessarily a result of genomic changes, it’s more like to be a consequence of altered biotic and abiotic conditions changing the expression of genes.

Line 320-324 This analysis should be presented in the results. Also the r^2 values are very low suggesting that if density dependent selection is occurring it's explaining less than 10% of the variation in growth rates.

Line 338 This context about the study site should be provided much earlier in the manuscript because it's highly relevant to understanding the mechanisms of T and eCO₂ influence the microbes by changing plant processes and soil resources.

Line 345 -354 This paragraph is much more helpful than many of the previous. I suggest the authors expand this discussion and connect it better to the growth responses.

Line 355 Resources are always limiting, perhaps it would be better to highlight the stoichiometric imbalance?

Line 442 Suggest change to "Incubation conditions were similar to those previously described". So it's clear it's not the same experiment.

Reviewer #3 (Remarks to the Author):

Summary. The increase of greenhouse gas emissions give rise to a global warming effect that ultimately influences both above-ground and below-ground processes across Earth's ecosystems. Microbial communities, which underpin the biogeochemical processes that often shape ecosystem function, are no exception to this, and although research shows they can adapt to this warming, population dynamics and ecophysiological responses to these disturbances are not currently known. The manuscript reviewed titled "Phylogenetic constraint and environmental acclimation link to soil" set out to describe an integrated, trait-based understanding of these dynamics with their data which includes 16S rRNA sequences, and qSIP.

This research evaluates two fundamental hypotheses, that were clearly outlined in the introduction and will be referred to for the rest of this document as H1 and H2.

H1: Microbial growth is phylogenetically conserved in response to nutrient pulses following rewetting events.

H2: Warming shifts pre-existing life strategies, which would conflict with phylogenetic signal.

Overall- the manuscript was not challenging to read and except for a few spaces I noted below, I could track the framing quite easily without a sizable background in this specific research area. I appreciated the linkage of climate perturbations in the field to wetting events in the lab, especially the use of growth accelerated and growth delayed. There were 4 areas where additional clarification was warranted. These comments are addressed below in more detail but broadly summarized here.

(1) The hypothesis testing framework doesn't seem robustly evaluated in the text as written.

(2) Novelty of this study relative to other literature – While I recognize after skimming these other prior papers (two by Firestone lab, one by Blasewitz is quite similar) there are novel areas in this new work, this needs to be more clearly reinforced by the authors, especially for a broader readership like in Nature Communications, to appreciate the relevance of this research.

More minor, but added up over time, (3) experimental design considerations and (4) language precision with necessary caveats.

[1] Testing of the hypotheses as currently framed and with data reported in the results.

a. I didn't see data that supports the hypotheses as they are written. It thus reads like the story is based on assumptions of processes happening, not actual measurements.

For example for H1, "Microbial growth is phylogenetically conserved in response to nutrient pulses following rewetting events" I would expect to see measurements of nutrient pulses. There is from what I can see no measurement of the nutrient pulse (DOC, TN, TC, CO₂ fluxes) as other studies on this

topic have done. Thus the authors seem to assume the mechanism without showing it. I would also expect to see metric of rewetting which would have initial, final water content. I note both these types of data are in used in Blacewicz, but also it seems glaring given how strongly the hypothesis was framed. Key data is missing that would be necessary to support this hypothesis as written.

b. For H2- what is meant by life strategy? I didn't see this defined in the introduction, nor in the results where this hypothesis was evaluated. Can the authors add more clarity/specifics here-How was life strategy explicitly defined here, how was it measured, what was the data?

c. Stylistic suggestion, remove italics on the hypotheses in the intro.

d. Can the authors put the timing of the original field experiment in the methods and or figure 1 experimental design, apologies if I missed it (line 439 in methods and line 85 intro discusses long term but I have no idea how this was done by content in this manuscript). This is important for H2 evaluation and only in line 126 do I infer that experiment that these soils were derived from was a decade? Please add this text clearly to fig 1 and methods so readers do not have to find another journal to understand basics of this story nor wait until results to understand experimental design (also apologies again if I missed it!).

e. Lastly given the authors go to great lengths to call out their hypotheses (italics) I wanted to the results to better link to these in the text. On revision I suggest more clear links would be helpful as it is clear I had a tough time tracking exactly what the authors were testing to support these claims.

[2] Novelty of this work relative to prior work.

I am loosely familiar with work that is cited in this manuscript (Placella et al) that evaluated a similar concept in grassland soils, published in PNAS 2012. "Rainfall induced CO₂ pulses result from sequential resuscitation of phylogenetically clustered microbial groups". This research aligns very closely with this study and uses nearly identical methods (qSIP), "taxon specific microbial growth and mortality patterns reveal distinct temporal population responses to rewetting in California grassland soils" Blacewicz et al, 2020. I found myself wondering what sets this work apart from the prior work and the new insights that were provided here. I think the final take-away and why we should care also could be better communicated for relevance to this journals readership.

[3] Experimental design

(1) I have some concerns about the experimental set up. From what I read in figure 1 I did not see a pre-wet treatment (please confirm this), everything was compared to samples collected after the treatment. As cited in the intro, and other papers have all shown, a rapid response is already observed within 1 hour thus growth rate needs to be measured from prewet condition. How long is "immediately"- was the entire experiment set up and then samples removed after? If accurate, address concerns with experimental design, or if I missed it add more clarity to Fig 1. I believe this first since not a true control should be T1 (after treatment) as T0 implies before effect and the papers cite a rapid effect following this treatment. This comes important right away as I found myself wondering if on line 122 the differences in richness were due to wetting event or prior historical artifact.

Note prior study by Blacewicz that authors reference often used these treatments and responder categories.

- a. Prewet {important control}
- b. Primary 3 hours
- c. Secondary 24 hours
- d. Tertiary 72 hours
- e. Delayed 168 hours

(2) Prior studies used fresh soils- collected within a week of incubation. However this study used soils that had been stored in the fridge or at 4C. For how long prior to the study being done- I was confused by language at the end of line 435 and the beginning of line 443 (fresh). Why were the soils sieved as that disrupts structural features?

(3) Line 508: What version of RDP Classifier was used?

(4) Line 511: The accession numbers of the deposited 16S rRNA data is stated as "not available until 2023/06/01". This needs to be updated on the resubmission as we could not download the data.

(5) Line 159 The authors said they used same growth classification as in Blacewicz (which I

appreciated linking to prior papers), but I didn't see that manuscript used language of rapid, intermediate, or slow. In fact I couldn't find intermediate in that manuscript – on a quick search. This becomes an issue and I found myself unsure if their calls for “slow / medium / fast” responders are appropriate as this wasn't entirely easy for me to follow. For example, looking at Figure 2, I can see that abundances for some of the groups are very similar across all days, and so calling an organism as “rapid” because it is ever so slightly higher in sample days 0-1 than in sample days 0-6 seems imprecise. However, when they show Fig 3 and Z-Score those values I can definitely see some distinctions by the growth rate categories. Can the authors clarify how this was done here, as this is important to understand so figures/results can be interpreted.

[4] Language precision

(1) Mainly, the link between microbial taxonomy and microbial metabolism is known to not be straightforward, especially in diverse microbial communities like soils that lack good reference genome databases. I do appreciate that the authors introduce caveats to the end of discussion paragraph 2 very well – and so I am aware they know the limitations of their methods. However, it would be helpful to start framing those caveats earlier within discussion paragraph 1 as well as throughout the results section when the piecrust was introduced. For context, as a reader who does metabolisms in soils I found myself somewhat disappointed when piecrust data was introduced and not properly caveated. Particularly, it might be helpful to introduce briefly at lines 226-236 in the results and lines ~278. These caveats are necessary to not potentially overstate the author's findings, and to make sure the reader knows the authors understand the very clear limitations of these methods.

<https://microbiomejournal.biomedcentral.com/articles/10.1186/s40168-020-00815-y>

With this approach, we found reasonable performance for human datasets, with the metagenome prediction tools performing better for inference on genes related to “housekeeping” functions. However, their performance degraded sharply outside of human datasets when used for inference. Also more: <https://academic.oup.com/gigascience/article/doi/10.1093/gigascience/giab090/6505123>

(2) In the discussion section there must be some additional caveats introduced to some of the claims the authors are making. Some of the language that is used within discussion paragraphs require data that the authors do not have (e.g., whole-genome sequencing or metagenomics), or that are somewhat contradictory to their results (e.g., “we identified major forces of variation” when only 20% of the data is explained by these metrics). As such, softening of the language is recommended- lines are noted below- and these will not adjust the outcomes of this study, but support more precise interpretation.

Line 274-275: Please be mindful and introduce some caveats to the statements made on these lines. What else other than phylogeny and “core genes” could possibly drive these differences?

Line 330: Consider removing this final sentence The explanation in the upcoming paragraphs is both helpful and needed prior to making the (valid!) claim you are making on line 352.

Line 366-368: Careful with the phrase “...we identified the major forces...”. Changing this to “we identified some of forces...” is prudent given we do not know all factors. Specifically, there is less than 20% variation explained by these variables in both the rapid and intermediate responders. In other words, 80-89%% of the variation is not explained by these data, and you cannot call them major drivers. Further, please add a strong caveat into this section that speaks to what the other 90% of variability in Rapid and Intermediate responders might relate to, as it is clearly not phylogenetic or environmental acclimation.

Line 374: Consider softening this language. There are possibilities not accounted for here. “...consisting of numerous proteins, evolved more slowly...” with “...consisting of numerous proteins, usually evolve more slowly...”

Line 377: Consider softening this language. Change “...microbial trait of rapid growth was inherited vertically...” to “...microbial trait of rapid growth was likely inherited vertically...”.

Line 389-390: The claim that they are governed by specific environments is somewhat misleading. They are primarily governed by environmental variables that may or may not be conserved across an environment. Please rewrite this sentence to reflect that.

Line 392: Consider adding in "...the significance of phylogeny and their respective functional traits..."

Line 393: Per your results, this was not always the case. Please soften this language to reflect your results. "...response to wet-up events are mostly phylogenetically conserved..."

Line 398-399: This claim needs a citation – as you do not have “global” evidence for this. Alternatively, the claim needs to soften the language. "...within a given species locally, and, potentially, globally."

[5] Specific line comments:

please not just 16S all throughout the manuscript- it should be “16S rRNA gene” to signal primers were used to amplify this gene or similarly “16S rRNA amplicon”.

Line 178 tone down claim- suggesting that “under the conditions evaluated here” vertical inheritance was essential for the distribution of growth rate traits in response to rewetting. I think functional traits is a bit too broad.

Line 60: I believe maybe the intended word was “mediated” instead of “medicated”?

Line 111: Consider making this title reflect the major result of the section as you did with the rest of the titles. It helps the reader focus on the message.

Line 133: according to (6). Consider just writing to “according to a previous publication (6)” as the formatting will look better.

Line 159: Change “The phylogenetic tree...” to “A phylogenetic tree...” and consider joining the sentence after for clarity. “A phylogenetic tree including all responders ... was constructed, and 3 phylogenetic indices were used to estimate the ...”.

Line 170: Consider adding the word additional to “We used two additional indices...”, as you mention above you have 3, and these are the other two.

Lines 226-236: Please add a sentence here regarding the caveat that microbial phylogeny is not always representative of microbial metabolism.

Line 239: Are you trying to say that all categories, with the exception of intermediate responders in eT, were clustered at phylogenetic branches? If so, please consider writing this sentence like that to enhance clarity.

Line 247: Consider changing the word “expresses” to “exhibits”.

Line 247: Consider adding the word ultimately to: “...that a species expresses, which ultimately weakened the strength of phylogenetic patterns.” to make your closing statement more powerful.

Line 281: How many are “most” members? It would be helpful to add in exactly how many.

Line 315: The tense for this should read: “which is ubiquitous in the microbiome and considerably shapes the structure and functions...”. Style wise I generally avoid the term ubiquitous as it means in everything and since we cannot measure to exhaustion in microbes...but that is my thing I realize.

Line 325-328: Please reword this so that it is understood that these findings are from another study.

Maybe start the sentence with “A general concept is that these..., and previous works have shown that under resource limitation, per capita resource...”.

Line 332: Consider removing “and elevated CO2 concentration” from this first sentence, as this paragraph is on warming specifically, and the following one is about CO2 concentration.

Line 345: Consider adding in the removed “and elevated CO2 concentration” from the comment above here. This then frames this paragraph as the CO2 paragraph.

Line 370: Add the word “and” to “...former, AND slow responders by the latter” for clarity.

Lines 373-378: Careful with the statements in these last few sentences. Some caveats and softening of language here are necessary. The growth-related genes do not need to be located on the core genome, as the referenced paper (27) states. It is possible it is related to environmental adaptations, and some phylogenies are more prone to that than others due to a myriad of factors. Softening the tone with words like “tend to be located” instead of “are located” (line 374).

Line 541: What happened with the taxa that did not fit into the 3 delimited strategies? Were they removed? Kept? Please elaborate here.

Figure 6: The labels for “intermediate” say “intermediated” in some instances. Please change to keep consistent.

Figure S3: Change “birth rates” to “growth rates” to keep consistent with axes label. Also – the legend seems to be missing from this supplementary figure. Please add. Also – from the manuscript text: “Taxa with fast growth rates were conserved within several 143 bacterial phyla (Fig. 2 and Fig. S3)”. Could you label these within Figure S3 as you did for Figure 2? What are the smaller bar plots inside some of these bar plots?

Figure S5: This figure also does not contain a figure legend. Please add.

Point-to-point Responses to the referees' comments:

Reviewer #1 (Remarks to the Author):

The authors determined microbial population-specific growth rates by ^{18}O -DNA quantitative stable isotope probing (^{18}O -qSIP) in soils subjected to a free-air CO_2 enrichment (FACE) combined with warming. The soils were sampled from the FACE experiment in Kangbo village Guli Township, Changshu municipality in Jiangsu Province, China, and the experiment started in 2010. The authors found that bacterial taxa could be separated into three distinct groups with different growth strategies which were phylogenetically conserved in soils that were not manipulated with higher CO_2 or temperature environments but when these manipulations were implemented the phylogenetic signal disappeared.

The work appears to be well and carefully done and the data and analyses are of great value to the larger microbial ecological community. There are very few studies in which the growth rates of bacterial populations from FACE experiments have been investigated via qSIP and the large quantity of data that comprises the study described in this manuscript, will substantially advance our knowledge of bacterial responses to elevated CO_2 and temperature manipulations. However, I think the manuscript will require major changes and a considerable rewrite.

Response: We appreciate the encouragement and positive comments. We have followed the suggestion to considerably rewrite the manuscript in the Introduction, Result and Discussion sections, shown in our point-to-point responses below.

My chief complaint about the manuscript is that it is too long because of redundant information. For instance, L114 describes the soils that were collected in the results section, it is again described in L432 and in the legends of various figures.

Response: We have carefully gone through the manuscript to remove the redundant sentences. We have also sought help from native English speakers to polish the language. We apologize for not listing all of those changes in this response letter since

there are too many changes, which are marked in red in the revised manuscript. Using the example pointed out by the reviewer, we have only retained the information about soil collection at the beginning of the Results section (Line 116-118) after deleting similar sentences elsewhere in the main text and figure legend.

Similarly, the discussion section has many sentences that restate findings presented in the Results section. Another example is the description of growth strategies (L192 to 204) which occurs repeatedly in the manuscript. It is important to write a concise manuscript in which information is only presented once.

Response: Since the comment is related to the preceding one, please see our response above. For the example of growth strategies (L192 to 204 in the original version), we have kept the description in the legend of Fig. 4 (Line 760-764) and deleted similar sentences elsewhere.

I also find that the discussion section is too long and while it contains presentation of other studies, these findings are often not well related to the growth rates of bacteria measured by the authors. For instance, the authors did not measure the number of rRNA genes on a taxa's genome and so there appears limited need to extensively discuss relationships between rRNA copy number and growth rates. If there are difficulties relating other studies to the authors findings, the discussion of those studies should be removed from the paper.

Response: We have substantially shortened the Discussion by deleting the contents of other studies that have little relevance to the growth rates in the discussion section, including the discussion about the relationship between rRNA copy number and growth rates, and the discussion about the variations of genome caused by climate change.

Below are some, but not all, examples that have been deleted in the Discussion,

1) “Members of the class Bacilli generally have high copy numbers of rRNA operons, enabling them to produce ribosomes rapidly and assimilate substrates immediately following nutrient addition”;

- 2) “In addition, the extremely high intraspecific variation of the genome also promotes such fast adaptations”;
- 3) “Finally, these shifts in life strategies within species may have emerged through horizontal dissemination of genes, which is ubiquitous in the microbiome and considerably shaped the structure and functions of global ecosystems”;
- 4) “Slow responders, with low maximum growth rates and low rRNA operon copy number, are thought to have growth advantages under limiting resource conditions”.

I think the introduction focuses too much on the “Birch effect”. While soil wetup effects are well documented, they always occur when liquids, be it water or a glucose solution, are added to soil. I think in soils that routinely dry and wet back up, unlike soils from Mediterranean climates such as California, have minimal Birch effects upon rewetting. The paper is not really about the “Birch effect”, it is about the impact of CO₂ and temperature on bacterial growth rates in soil and the introduction should focus on that.

Response: We have followed the suggestion to minimize the introduction of "Birch effect". As a result, we have deleted the sentences “This phenomenon is known as the birch effect”, and deleted the phrases “response to soil wet up” and “to the resource pulse” to better focus on climate effects (For example, Line 89-91).

We have measured the CO₂ production rates during the incubation according to the suggestion of the reviewer #3. Compared with the previous study in Californian soils (Placella et al. 2012), there were lower CO₂ production rate and longer time for reaching peak CO₂ production rate in our experiment, verifying the reviewer' prediction that the soil used in our study has minimal Birch effects upon rewetting.

Reference:

Placella et al. (2012) Rainfall-induced carbon dioxide pulses result from sequential resuscitation of phylogenetically clustered microbial groups. PNAS 109 (27), 10931

I suggest changing the title of the manuscript from “Phylogenetic constraint and environmental acclimation link to soil microbial growth strategies under CO₂

enrichment and warming.” To something like: “Elevated temperature and CO₂ strongly affect growth strategies of soil bacteria.”

Response: We have followed the suggestion to change the title to “Elevated temperature and CO₂ strongly affect the growth strategies of soil bacteria”.

More minor edit suggestions:

Abstract: I suggest deleting the first few sentences of the abstract and start with the sentence in Line 25.

Response: Done. Please check the revised version in Line 22.

L135 insert “the” before “6”

Response: Done (Line 134).

L43 Delete “The evidence is overwhelming that”

Response: Done.

L159 change “the” to “a” before phylogenetic

Response: Done (Line 168).

L217 change “shared with three comparisons” to “shared among the three treatments”

Response: Thanks. Sorry for the unclear statement. Based on your suggestion, we have revised this sentence to “shared among the three treatment comparisons” (Line 223).

I struggle with the following argument: “The essential genes for microbial growth are considered to be the core genome, which is generally inherited vertically and reflects phylogeny strictly. This helps explain the observed phylogenetic conservation of the strategies related to growth response.” How do the authors know that the core genome allows microorganisms to grow faster or slower (i.e. be more competitive)?

Response: Liu et al. (2012) showed that the growth-related essential genes are mainly belong to the core genome, which is strongly influenced by phylogeny. This explains why phylogenetic signals were detected for all three growth responders (Table 1). To clarify it, we have revised this sentence to “The growth-related genes mainly belong to the core genome, which is mostly inherited vertically and reflects the phylogeny (Tamames et al. 2016). This explains the phylogenetic signals of the three growth strategies (Table 1), which supports the first hypothesis that phylogeny influences the growth strategies of bacterial species after soil wet-up” (Line 270-274).

References:

Liu et al. (2012) Comparative genomics of Mycoplasma: analysis of conserved essential genes and diversity of the pan-genome. PLoS One 7(4), e35698

Tamames et al. (2016) Quantifying the relative importance of phylogeny and environmental preferences as drivers of gene content in prokaryotic microorganisms. Frontiers in Microbiology 7.

L280 Suggest change of “visually evident based” to “evident”.

Response: Done (Line 280).

L293 Suggest changing “validated” to “supported”

Response: Done (Line 273).

L295 Suggest deleting “This helps grasp the relationship between phenotypic characteristics and the phylogeny of microorganisms.

Response: Deleted.

L 309 Does the sentence “Microbes possessed metabolic plasticity for carbon use to suit resource availability or interspecies interactions, which promoted or reduced microbial growth^{28, 29, 30} refer to microorganisms in this study? If not, that should be clearly indicated.

Response: As this sentence is dispensable, we have deleted it from the Discussion.

L310 “extremely high intraspecific variation of the genome” It is unclear what genome this sentence refers to

Response: As we provided no evidences for the variations of the genome (a comment from reviewer #2), we have deleted this sentence in the discussion section.

L355 suggest changing “for resources under limitation” to “for limited resources”.

Response: Done (Line 327).

L357 suggest changing “mitigated” to “mitigating”

Response: Done (Line 330).

L370 suggest changing “microorganismic” to ”microbial”.

Response: Done (Line 341).

L442 “The incubations were described in detail previously⁶.” I think this refers to another study and not the actual incubations in this study.

Response: We have revised this sentence into “The incubation conditions were similar to those reported in a previous study (Blazewicz et al. 2020)” (Line 396).

References:

Blazewicz et al. (2020) Taxon-specific microbial growth and mortality patterns reveal distinct temporal population responses to rewetting in a California grassland soil. *The ISME Journal* 14, 1520–1532

L513 suggest deleting “As ¹⁸O labeling occurs during cell growth via DNA replication”

Response: Deleted.

Figures:

I am unclear what “low growth” means in bacterial phyla list of Figure 2

Response: The “low growth” refers to the bacterial phyla with low growth rates.

To clarify it, we have replaced “low growth” with “Others”, and revised the legend of Fig. 2: “Bar color: bacterial phylum or class (10 of them are shown, accounting for > 93% of growth rates, and remaining phyla are designated as “Other”)” (Line 745-746).

Figure 5 The results presented in Figure 5 are described in the text and do not require an entire figure.

Response: Since it is not a very important figure, we have moved it to supplemental materials (i.e., Fig. S6 in the revision).

Figure 6 seems to recap information from other figures and could also be removed from the manuscript

Response: We have removed Fig. 6 from the manuscript.

Reviewer #2 (Remarks to the Author):

The manuscript by Ruan et al. describes the growth responses of soil bacteria following rewetting in soils from a long-term global change experiment. The authors used qSIP with ^{18}O water to measure microbial growth rates after 1, 3, and 6 days. Based on the day of maximum growth taxa were classified as “rapid”, “intermediate” or “slow” responders. In general, the experimental design and methods appear to be of sufficient quality. The results are likely to be of interest to the microbial ecology community and may help advance our understand of how microbial systems will respond to global change.

I do have some significant concerns about some of the data analyses, the framing of the paper, and the interpretation of the results.

Response: Thanks for your encouragement! We have followed your suggestions to considerably rewrite the manuscript, shown in our point-to-point responses below.

1) Why are the treatments grouped together for the phylogenetic dispersion, Blomberg’s K, and Pagel’s lambda analysis but not the NTI analysis?

Response: We have added the NTI analysis in the revised manuscript (Table 1). Both rapid and slow responders exhibited phylogenetic clustering ($p < 0.001$), consistent with the results of other phylogenetic analyses (Table 1). However, only the intermediate responders had a random distribution on the phylogenetic tree ($p > 0.05$). This difference may be due to the different algorithms. We have added the description of NTI in Line 186-187 of the revised manuscript as “The nearest taxon index (NTI) showed a clustered distribution of rapid and slow responders at the end of the branches of the phylogenetic tree”.

Table 1 Phylogenetic indices of three growth strategies

	Phylogenetic dispersion	Blomberg's K	Pagel's λ	NTI
Rapid	0.38 ***	0.16 ***	0.59 ***	3.18 ***
Intermediate	0.59 ***	0.12 **	0.40 ***	0.06
Slow	0.77 ***	0.11 **	0.24 ***	4.11 ***

The smaller D values represent more clustered phylogenetic patterns, while the smaller K and λ indicate more random phylogenetic patterns. The p values were calculated with 1000 permutations based on trait prevalence and phylogeny. Significance levels: ***, $p < 0.001$; **, $p < 0.01$; *, $p < 0.05$.

For the analyses in table 1 how did you treat OTUs that were present in more than one treatment and had differing responses? Were the data converted into binary (1s and 0s) before all the phylogenetic analyses? These details should be explained in the manuscript.

Response: Yes, the data were converted into binary (1s and 0s) before all the phylogenetic analyses. For the OTUs that were present in more than one treatment and had differing responses, we performed the following processing: an OTU was classified as a corresponding growth responder when it exhibited a certain growth strategy in any treatment. For the phylogenetic analyses of three growth strategies (combined four climate treatments), 1) We obtained the 0 / 1 matrix based on the distribution of the responder in four climate treatments (a matrix with four columns of data); 2) We identified one OTU as a “potential” responder based on its growth response in any treatment. As a result, an OTU can be divided into one or more kinds of “potential” growth responders. Among 1017 OTUs, 53% OTUs were divided into only one kind of growth strategies, 45% OTUs have two, and less than 2% of OTUs were classified into all three types of growth strategies.

We have added the details in the Methods section “The data were converted into binary matrices (1s and 0s) before all phylogenetic analyses. For the OTUs that were present in more than one treatment and had differing responses (in the analyses when all climate treatments are combined, i.e., in Table 1 and Fig. S6), the OTU was classified to the respective growth responder when that taxa exhibited that specific growth strategy in any treatment” (Line 520-525).

2) The authors frequently conflate the categorized growth responses they observed with ecological or life strategies. This study focuses on one trait the timing of maximum growth after rewetting. One trait cannot be used to assign an ecological strategy as ecological strategies are defined by a suite of traits that covary due to eco-evolutionary tradeoffs.

Response: We have replaced the “life strategy” or “ecological strategy” with the “growth strategy” since we focused on microbial growth dynamics throughout the text.

3) The authors seem to confuse adaptation, acclimation, and plasticity in their interpretation of changes in the timing of maximum growth following rewetting.

Response: To our knowledge, 1) acclimation has been defined as the modification of physiological or biochemical traits within an individual organism in response to changes in environmental variables in a laboratory or field setting (Wilson and Franklin 2002); 2) Adaptation is also the physiological modification occurring within a species that result from chronic exposure to naturally occurring environmental challenges (over several generations) that facilitate an enhanced ability to survive and reproduce in a particular environment. This means that adaptation includes changes in both phenotypic and genetic traits; 3) Phenotypic plasticity occurs when a genotype expresses different phenotypes in different environments (Turcotte et al. 2016).

Therefore, we have modified the terminology throughout the text. We think that “acclimation” fits well in most cases. And this terminology also meets your suggestion in the following comment.

References:

- Wilson RS, Franklin CE. (2002) Testing the beneficial acclimation hypothesis. *Trends in Ecology & Evolution* 17(2), 66-70;
- Turcotte MM et al. (2016) Phenotypic Plasticity and Species Coexistence. *Trends in Ecology & Evolution* 31(10), 803-813.

As with “ecological strategy” these terms are too often used and abused in the field of microbial ecology perhaps contributing to their misuse in this paper.

Response: To reduce the too often use of ecological strategy, we have replaced the “life strategy” or “ecological strategy” with the “growth strategy”.

They suggest that communities and OTUs have adapted to the global change factors via changes in their genome but provide no evidence for this.

Response: Thanks for your suggestion. Since this statement is not so accurate and it is not an indispensable sentence, we have deleted the sentences about “microbial acclimation to climate via changes in microbial genome” in the manuscript. In addition, we removed the word “genomes and” from the original version and rephrased the sentences (check Line 286 and Line 337); We also deleted similar sentences “In addition, the extremely high intraspecific variation of the genome also promotes such fast adaptations” and “Finally, these shifts in life strategies within species may have emerged through horizontal dissemination of genes ...” from the original version.

It’s far more likely that the global change factors altered the soil environment by changing plant and microbial processes that impact water, carbon, and nitrogen availability) which in turn impact the timing of microbial growth following soil rewetting. Changes in the trait (timing of growth) in response to global change factors indicate phenotypic plasticity that enables the organisms to acclimate to the altered environment. Unfortunately, the authors provide little insight into how changes in the soil could be mediating.

Response: Agreed. Prior studies in this FACE experimental station suggested that soil abiotic and biotic conditions were influenced by warming and elevated CO₂ concentration, including decreased soil pH and increased organic carbon content (Liu et al. 2014; Liu et al. 2021; Xiong et al. 2019). Moreover, plant physiological traits (e.g. total biomass, grain yield, and gross photosynthetic rate) were significantly decreased in warming treatment, and increased by elevated CO₂ concentration (Cai et al. 2016; Wang et al. 2016). We have added relevant description in the result section (Line 105-109): “Based on this FACE experiment, elevated temperature and CO₂ concentration altered plant physiological traits: total biomass, grain yield, and gross photosynthetic

rate decreased under warming but increased under elevated CO₂ concentration (Cai et al. 2016; Wang et al. 2016). Soil pH decreased and organic carbon content increased in all three treatments (Liu et al. 2021)”.

Recently, we have re-collected the soil samples from the FACE experimental station in July 2022, and measured soil chemical properties. Consistent with the results of previous studies, long-term climate changes have significantly decreased soil pH and increased organic carbon content (please see Table for the response letter).

Table for the response letter Soil properties in the FACE field experiment station

Treatment	pH	AP (mg · kg ⁻¹)	AK (mg · kg ⁻¹)	SOC (g · kg ⁻¹)	TC (g · kg ⁻¹)	TN (g · kg ⁻¹)
Contr	7.13±0.08 a	29.71±14.03 a	23.00±1.73 a	13.69±3.35 b	14.90±3.00 a	2.28±0.31 a
eT	6.91±0.08 b	30.11±4.22 a	24.33±1.53 a	16.97±1.17 ab	17.87±1.43 a	2.59±0.23 a
eCO ₂	6.89±0.10 b	23.50±5.95 a	22.00±2.65 a	16.37±3.77 ab	17.89±2.55 a	2.30±0.45 a
eTeCO ₂	6.86±0.06 b	28.57±14.39 a	25.33±4.93 a	19.06±0.26 a	24.45±6.56 a	2.90±0.71 a

Values represent mean and standard deviation (n=3). Contr represents the ambient temperature and CO₂ concentration; eT represents warming only; eCO₂ represents elevated CO₂ concentration only; and eTeCO₂ represents combined warming and CO₂ enrichment. Different letters indicate significant differences between treatments

According to the suggestion by reviewer #3, we measured soil chemical metrics (DOC, DN, TN, and TC) at different incubation time points (Fig. S1). Further, we performed RDA to explore the driving factors of microbial growth dynamics (Figure for response). We found that soil DOC content could explain the variations of bacterial growth at different incubation intervals. These results indicated that soil conditions influenced microbial growth dynamics.

Figure for the response letter

Redundancy analysis (RDA) was performed to explore the explanatory factors for the variations of bacterial growth among three incubation intervals by using soil biochemical traits. We performed the function “anova” in vegan (R package) to assess the significance of each factor. The solid arrows represent the p values of environmental factors less than 0.05 (permutations = 999), while the dashed arrows do not ($p > 0.05$).

References:

- Liu et al. (2014) Short-term responses of microbial community and functioning to experimental CO₂ enrichment and warming in a Chinese paddy field. *Soil Biology and Biochemistry* 77, 58–68;
- Liu et al. (2021) Long-term elevated CO₂ and warming enhance microbial necromass carbon accumulation in a paddy soil. *Biology and Fertility of Soils* 57(5), 673-684;
- Xiong et al. (2019) Molecular changes of soil organic matter induced by root exudates in a rice paddy under CO₂ enrichment and warming of canopy air. *Soil Biology and Biochemistry* 137, 107544
- Cai et al. (2016) Responses of wheat and rice to factorial combinations of ambient and elevated CO₂ and temperature in FACE experiments. *Global Change Biology* 22(2), 856-874
- Wang et al. (2016) Size and variability of crop productivity both impacted by CO₂ enrichment and warming—A case study of 4 year field experiment in a Chinese paddy. *Agriculture, Ecosystems & Environment* 221, 40-49

the impacts of eCO₂ and temperature on the timing of microbial growth. At a minimum the environmental data used for the analysis presented in Figure 5 should be included in the manuscript.

Response: Figure 5 has become Fig. S6 in the revision, based on the suggestion from Reviewer #1. The environmental data used for its analysis are three distribution matrices (binary) of three growth responders in the four climate treatments.

We have added this description in the methods section as “The distribution matrices (binary) of three growth responders in the four climate treatments were used to represent the impact of CO₂ and temperature on bacterial growth” (Line 530-532).

4) The analysis in Fig 5, what exactly is being predicted? What is “y” ? Is it a 1 or 0 indicating the membership in a response category? Or is it the growth rate if the OTU is in the response category?

Response: The analysis in Fig 5 (now Fig. S6, according to the suggestion of the reviewer #1) aimed to predict the membership in a response category with the amount of variance explained by microbial phylogeny and environmental acclimation.

Yes, the “y” is a 1 or 0 indicating the membership in a response category. We have added the descriptions of these details in the manuscript, “To predict the microbial growth strategy (i.e., “y”, a 1 or 0 indicating the membership in a response category)” (Line 532-533).

Specific comments

Introduction – The introduction does not adequately establish why understanding the influence of phylogeny on function is important.

Response: We have added sentences to elaborate the significance of phylogeny on the trait-based studies as “Microbial phylogeny offers a framework to clarify the response mechanisms to specific environments and, consequently, the particular functions of

microbial taxa within an ecosystem (Martiny et al. 2015; Philippot et al. 2010). The phylogeny is crucial for multiple phenotypic and functional characteristics of microbes. These include pH and salinity tolerance (Martiny et al. 2015), the utilization of various organic substrates (Barnett et al. 2021), and the metabolic as well as growth response to pulse events (Morrissey et al. 2016; Placella et al. 2012), with consequences for the overall community assembly” (Line 58-64).

References:

- Martiny et al. (2015) Microbiomes in light of traits: A phylogenetic perspective. *Science* 350(6261), aac9323;
- Philippot et al. (2010) The ecological coherence of high bacterial taxonomic ranks. *Nature Reviews Microbiology* 8(7), 523-529
- Barnett et al. (2021) Multisubstrate DNA stable isotope probing reveals guild structure of bacteria that mediate soil carbon cycling. *Proceedings of the National Academy of Sciences* 118(47), e2115292118
- Morrissey et al. (2016) Phylogenetic organization of bacterial activity. *The ISME Journal* 10(9), 2336-2340;
- Placella et al. (2012) Rainfall-induced carbon dioxide pulses result from sequential resuscitation of phylogenetically clustered microbial groups. *Proceedings of the National Academy of Sciences* 109(27), 10931

Line 126 – I suggest you change “adapted” to “acclimated”

Response: We have followed the suggestion (Line 125).

Line 142 – Remove “conserved”

Response: Done (Line 142).

Line 144 – Fig 2 shows proportions but is labeled as (%). The percentages are a little confusing. I think it would help if you reminded the reader how to interpret them. For instance, 29% of new 16S rRNA gene copies.

Response: We have revised it to “29% of new 16S rRNA gene copies” (Line 143-144).

Line 172 This is incorrect – Blomberg’s K can be greater than 1. Make sure you fully understand the statistics you are employing.

Response: Yes, the values of Blomberg's K can be greater than 1 and the range of Pagel's λ is 0-1 (Münkemüller et al. 2012; Wang and Clarke 2014). We have revised this sentence to "In most cases, the range of Pagel's λ was 0-1 (from random distribution to strong phylogenetic signal), whereas Blomberg's K can be higher than 1, indicating stronger trait similarity between related species" (Line 180-183).

References:

- Münkemüller et al. (2012) How to measure and test phylogenetic signal. *Methods in Ecology and Evolution* 3(4), 743-756
- Wang and Clarke (2014) Phylogeny and forelimb disparity in waterbirds. *Evolution* 68(10), 2847-2860

Line 176 If a trait is conserved it should have a higher phylogenetic signal than expected under Brownian motion ($k > 1$). This is not the case for the responses observed, there is an influence of phylogeny but that does not indicate it's conserved.

Response: We have rephrased this sentence to "Collectively, bacterial growth responses to sudden moisture increase are driven by phylogeny" (Line 187-188).

Line 226-236 The Picrust analysis and its results here and the heat map in Fig 4c are not well integrated with the rest of the manuscript. They feel tacked on and do not meaningfully enhance the manuscript. I suggest they be removed as they detract from the focus of the paper.

Response: We have followed the suggestion to delete the contents of the Picrust functional prediction.

Line 237 This section would fit better with the other phylogenetic analysis.

Response: As suggested by the reviewer, we have calculated three phylogenetic indices, Phylogenetic dispersion, Blomberg's K and Pagel's λ , for three growth strategies under different climate change treatments (Table S1). We have added the description of these results in Line 240-244:

"Furthermore, the three additional phylogenetic indices – phylogenetic dispersion, Blomberg's K and Pagel's λ – were similar to the result of the NTI index, showing the

phylogenetic signals in most growth strategies (\$p < 0.05\$ ). The phylogenetic signals of rapid responders were weakened after climate change (Table S1)”.

Table S1 Phylogenetic indices of three ecological strategies across the treatments

Field treatment	Strategy	Num taxa	Phylogenetic dispersion	Blomberg's K	Pagel's λ
Contr	Rapid	107	0.157 ***	0.181 ***	0.623 ***
	Intermediate	453	0.834 ***	0.113 **	0.139 ***
	Slow	190	0.850 ***	0.094	0.144 ***
eT	Rapid	184	0.402 ***	0.142 ***	0.491 ***
	Intermediate	675	0.618 ***	0.129 ***	0.437 ***
	Slow	14	1.064	0.083	0.011
eCO ₂	Rapid	159	0.509 ***	0.129 ***	0.407 ***
	Intermediate	684	0.784 ***	0.109 **	0.238 ***
	Slow	13	0.955	0.087	<0.001
eTeCO ₂	Rapid	142	0.485 ***	0.131 **	0.406 ***
	Intermediate	241	0.842 ***	0.101 *	0.116 **
	Slow	287	0.771 ***	0.109 **	0.244 ***

Contr represents the ambient temperature and CO₂ concentration; eT represents warming only; eCO₂ represents elevated CO₂ concentration only; and eTeCO₂ represents combined warming and CO₂ enrichment. Values in bold: $p < 0.05$. Significance levels: ***, $p < 0.001$; **, $p < 0.01$; *, $p < 0.05$.

Line 294 “unifying life strategies” , this language is far too strong to described the responses you measured.

Response: We have replaced the “life strategy” by “growth strategy” throughout the text. We have rephrased this sentence to “which supports the first hypothesis that phylogeny influences the growth strategies of bacterial species after soil wet-up” (Line 273-274).

Line 307 You have no evidence that environmental section is acting on the trait you measured.

Response: Thank you for your comments. We have deleted this sentence.

Lines 311 – 315 I think this interpretation is too strong. There was not statistical analysis to show that rates were significantly higher or lower across the 3 time points thus while the categorization may have changed it’s unclear whether this is a truly meaningful

change in the timing of growth. Even if the time of growth did change, this is not necessarily a result of genomic changes, it's more like to be a consequence of altered biotic and abiotic conditions changing the expression of genes.

Response: Agreed. As a result, we have deleted this sentence.

Line 320-324 This analysis should be presented in the results. Also the r^2 values are very low suggesting that if density dependent selection is occurring it's explaining less than 10% of the variation in growth rates.

Response: We have added the description of this analysis in the result section: “We correlated the per-capita growth rates with the initial population density (data from unfractionated DNA samples) of OTUs to infer the density-dependence of bacterial growth, which can shed light on the importance of ecological processes in structuring microbial communities (Blazewicz et al. 2020; Vellend 2016). There was no correlation between the per-capita growth rates and initial population density in the control soil (line model: $p > 0.05$). By contrast, we recorded significant negative correlations under eT and eCO₂ scenarios ($R^2=0.037$ in eT, and 0.062 in eCO₂; $p < 0.01$, Fig. S5), indicating negative density-dependent selection. The negative relationships, however, disappeared by the combined manipulations of CO₂ and temperature ($p > 0.05$)” (Line 155-163).

Additionally, we have explicitly indicated the caveats about the low R^2 in eT and eCO₂ scenarios as “Note that the R^2 between species growth rates and densities in eT and eCO₂ treatments were low (0.037 and 0.062, respectively), pointing to other driving forces shaping microbial growth” (Line 299-301).

References:

- Blazewicz et al. (2020) Taxon-specific microbial growth and mortality patterns reveal distinct temporal population responses to rewetting in a California grassland soil. *The ISME Journal* 14, 1520–1532
- Vellend (2016) *The Theory of Ecological Communities* (MPB-57). Princeton University Press

Line 338 This context about the study site should be provided much earlier in the manuscript because it's highly relevant to understanding the mechanisms of T and eCO₂ influence the microbes by changing plant processes and soil resources.

Response: We had added this information of the FACE experiment at the beginning of the Results section as “Based on this FACE experiment, elevated temperature and CO₂ concentration altered plant physiological traits: total biomass, grain yield, and gross photosynthetic rate decreased under warming but increased under elevated CO₂ concentration (Cai et al. 2016; Wang et al. 2016). Soil pH decreased and organic carbon content increased in all three treatments (Liu et al. 2021)” (Line 105-109).

References:

- Wang et al. (2016) Size and variability of crop productivity both impacted by CO₂ enrichment and warming—A case study of 4 year field experiment in a Chinese paddy. *Agriculture, Ecosystems & Environment* 221, 40-49;
- Cai C et al. (2016) Responses of wheat and rice to factorial combinations of ambient and elevated CO₂ and temperature in FACE experiments. *Global Change Biology* 22, 856-874
- Liu et al. (2021) Long-term elevated CO₂ and warming enhance microbial necromass carbon accumulation in a paddy soil. *Biology and Fertility of Soils* 57, 673-684

Line 345 -354 This paragraph is much more helpful than many of the previous. I suggest the authors expand this discussion and connect it better the growth responses.

Response: We have followed the suggestion to expanded this discussion as “Consequently, the resource limitations caused by warming and CO₂ enrichment are opposite: stronger C limitation by warming but increased N limitation by CO₂ enrichment. Stoichiometric imbalance or stress limit microbial growth, requiring more investments to maintain essential metabolic functions, or leading to dormancy (Lennon and Jones 2011; Price and Sowers 2004). Species with higher population densities may be more strongly limited under resource-scarce conditions compared to species with lower population densities (Vellend 2016). This partly explains the negative density dependence in eT and eCO₂ scenarios.” (Line 317-324).

References:

Lennon and Jones (2011) Microbial seed banks: the ecological and evolutionary implications of dormancy. *Nature Reviews Microbiology* 9(2), 119-130

Price and Sowers (2004) Temperature dependence of metabolic rates for microbial growth, maintenance, and survival. *Proceedings of the National Academy of Sciences* 101(13), 4631-4636

Vellend (2016) *The Theory of Ecological Communities* (MPB-57). Princeton University Press

Line 355 Resources are always limiting, perhaps it would be better to highlight the stoichiometric imbalance?

Response: Agreed. We have revised this sentence to “The ability to grow rapidly is crucial in competing for limited resources, especially under stoichiometric imbalance” (Line 327-328).

Line 442 Suggest change to “Incubation conditions were similar to those previously described”. So it’s clear it’s not the same experiment.

Response: Done. We have changed this sentence to “The incubation conditions were similar to those reported in a previous study” (Line 396).

Reviewer #3 (Remarks to the Author):

Summary. The increase of greenhouse gas emissions give rise to a global warming effect that ultimately influences both above-ground and below-ground processes across Earth's ecosystems. Microbial communities, which underpin the biogeochemical processes that often shape ecosystem function, are no exception to this, and although research shows they can adapt to this warming, population dynamics and ecophysiological responses to these disturbances are not currently known. The manuscript reviewed titled "Phylogenetic constraint and environmental acclimation link to soil" set out to describe an integrated, trait-based understanding of these dynamics with their data which includes 16S rRNA sequences, and qSIP.

This research evaluates two fundamental hypotheses, that were clearly outlined in the introduction and will be referred to for the rest of this document as H1 and H2.

H1: Microbial growth is phylogenetically conserved in response to nutrient pulses following rewetting events.

H2: Warming shifts pre-existing life strategies, which would conflict with phylogenetic signal.

Overall- the manuscript was not challenging to read and except for a few spaces I noted below, I could track the framing quite easily without a sizable background in this specific research area. I appreciated the linkage of climate perturbations in the field to wetting events in the lab, especially the use of growth accelerated and growth delayed. There were 4 areas where additional clarification was warranted. These comments are addressed below in more detail but broadly summarized here.

(1) The hypothesis testing framework doesn't seem robustly evaluated in the text as written.

(2) Novelty of this study relative for other literature – While I recognize after skimming these other prior papers (two by Firestone lab, one by Blasewtiz is quite similar) there

are novel areas in this new work, this needs to be more clearly reinforced by the authors, especially for a broader readership like in Nature Communications, to appreciate the relevance of this research.

More minor, but added up over time, (3) experimental design considerations and (4) language precision with necessary caveats.

Response: We appreciate the reviewer's positive comments. We have supplemented more key data and made extensive revisions to make the hypothesis tested robustly, reinforce the novelty of our work and increase the readability of our paper. Below is our point-to-point response in more detail.

[1] Testing of the hypotheses as currently framed and with data reported in the results.

a. I didn't see data that supports the hypotheses as they are written. It thus reads like the story is based on assumptions of processes happening, not actual measurements.

For example for H1, "Microbial growth is phylogenetically conserved in response to nutrient pulses following rewetting events" I would expect to see measurements of nutrient pulses. There is from what I can see no measurement of the nutrient pulse (DOC, TN, TC, CO₂ fluxes) as other studies on this topic have done. Thus the authors seem to assume the mechanism without showing it. I would also expect to see metric of rewetting which would have initial, final water content. I note both these types of data are in used in Blacewicz, but also it seems glaring given how strongly the hypothesis was framed. Key data is missing that would be necessary to support this hypothesis as written.

Response: Thank you for your comments. As also mentioned by reviewer #1, soil wet-up effects have been well documented: Previous studies suggested that the sudden increase in soil water potential stimulated carbon mineralization (Birch HF 1958; Fierer and Schimel 2002; Unger et al. 2010) and altered nutrient cycling (Schimel et al. 1989). These C and N transformations are almost entirely mediated by soil microbes (Placella et al. 2012; Blazewicz et al. 2020).

To verify the nutrient pulse after soil rewetting, we have collected soil samples again, re-conducted the incubation experiments in the laboratory, and measured soil

biochemical metrics (DOC, DN, TN, TC, CO₂ fluxes and the hydrolase activity of fluorescein diacetate) at four incubation time points. We found that the rate of CO₂ production increased significantly after water addition and peaked at 9 h of incubation (Fig. S1). The FDA hydrolase activity increased significantly after water addition and peaked at 3d of incubation. The content of dissolved organic carbon (DOC) decreased significantly and dissolved nitrogen (DN) increased gradually along the 6-d incubation, indicating the decomposition of organic carbon and nitrogen accumulation. These results indicated that there were nutrient pulses after rewetting dry soil. We have added the description at the beginning of the Results section (Line 109-113).

Fig. S1. Dynamics of soil biochemical characteristics during incubation. The CO₂ production rates are from 7 incubation time points, i.e., 0.125 d (3 h), 0.25 d (6 h), 0.375 d (9 h), 0.5 d (12 h), 1 d, 3 d, and 6 d after water addition. Letters (black lowercase) represent significant differences between sampling times, while the colored lowercase letters indicate significant differences between climate treatments at the same time point.

The data of other measured characteristics are from 4 different incubation time points, i.e., before water addition (prewet soil), incubations for 1 d, 3 d, and 6 d, with destructive sampling. Letters (black lowercase): significant differences between sampling times. Error bars: standard deviation (SD). Significance levels: ***, $p < 0.001$; **, $p < 0.01$; *, $p < 0.05$; ns, not significant.

References:

- Birch HF (1958) The effect of soil drying on humus decomposition and nitrogen availability. *Plant Soil* 10, 9–31;
- Fierer Noah, Schimel Joshua P. (2002) Effects of drying–rewetting frequency on soil carbon and nitrogen transformations. *Soil Biology and Biochemistry* 34, 777-787;
- Unger et al. (2010) The influence of precipitation pulses on soil respiration – Assessing the “Birch effect” by stable carbon isotopes. *Soil Biology and Biochemistry* 42, 1800-1810;
- Schimel et al. (1989) Spatial and temporal effects on plant-microbial competition for inorganic nitrogen in a California annual grassland. *Soil Biology and Biochemistry* 21, 1059-1066;
- Placella et al. (2012) Rainfall-induced carbon dioxide pulses result from sequential resuscitation of phylogenetically clustered microbial groups. *Proceedings of the National Academy of Sciences* 109, 10931-10936;
- Blazewicz et al. (2020) Taxon-specific microbial growth and mortality patterns reveal distinct temporal population responses to rewetting in a California grassland soil. *The ISME Journal* 14, 1520–1532

b. For H2- what is meant by life strategy? I didn't see this defined in the introduction, nor in the results where this hypothesis was evaluated. Can the authors add more clarity/specifics here-How was life strategy explicitly defined here, how was it measured, what was the data?

Response: It is not appropriate to use a life strategy here because one trait (i.e., microbial growth dynamics in our study) cannot be used to assign a life strategy. Based on the suggestion from reviewer #2, we have changed the “life history” and “ecological strategies” in the text to “growth strategies”.

Microbial growth strategies refer to the growth dynamics throughout the whole incubation. Growth strategies were classified as rapid, intermediate, and slow responders based on the timing of the peak growth rates. The data were the growth rates of species (mean of three biological replicates) at three incubation intervals (0-1, 0-3, and 0-6 d). To estimate population growth rates, we adopted the qSIP technique with steps of incubating soils with ^{18}O -labeled water and using density gradient

centrifugation of extracted DNA, followed by quantitative PCR (qPCR) and sequencing of the 16S rRNA gene in the resulting density fractions to estimate—for each bacterial taxon—the magnitude of density shift resulting from incorporation of ^{18}O into DNA of newly formed cells. Using additional measurements of 16S rRNA gene abundance at the beginning of the incubation, a model of bacterial growth was fitted and the population growth parameter was estimated for all taxa present throughout the incubation (Koch et al. 2018).

We have added the description of growth strategies in the Introduction section as “Taxon-specific population growth rates were assessed in three time intervals by using ^{18}O incorporation within DNA molecules (Koch et al. 2018). Three species growth strategies – rapid, intermediate, and slow response – were categorized based on the timing of the maximal growth rates (Fig. 1D)” (Line 91-94). The dataset of growth rates for the 1017 species has been uploaded as Supplementary Data 1.

Reference:

Koch et al. (2018) Estimating taxon-specific population dynamics in diverse microbial communities. *Ecosphere* 9, e02090

c. Stylistic suggestion, remove italics on the hypotheses in the intro.

Response: Done. We have removed the italics on the hypotheses (Line 65-66; Line 79-81).

d. Can the authors put the timing of the original field experiment in the methods and or figure 1 experimental design, apologies if I missed it (line 439 in methods and line 85 intro discusses long term but I have no idea how this was done by content in this manuscript). This is important for H2 evaluation and only in line 126 do I infer that experiment that these soils were derived from was a decade? Please add this text clearly to fig 1 and methods so readers do not have to find another journal to understand basics of this story nor wait until results to understand experimental design (also apologies again if I missed it!).

Response: We have followed the suggestion to add the information about the timing of the field experiments: “The Free Air Carbon dioxide Enrichment (FACE) experiment was established in 2010” (Line 101-102), and “Soil samples for qSIP incubation were collected in June 2020, i.e., ~10 years after the start of climate change experiment” (Line 116-118) at the beginning of the results section, and “an open-air experimental field was started in 2010” in the legend of figure 1 (Line 732-733).

e. Lastly given the authors go to great lengths to call out their hypotheses (italics) I wanted to the results to better link to these in the text. On revision I suggest more clear links would be helpful as it is clear I had a tough time tracking exactly what the authors were testing to support these claims.

Response: To better link the hypothesis and the corresponding results, we have added the sentences in the discussion section: “This explains the phylogenetic signals of the three growth strategies (Table 1), which supports the first hypothesis that phylogeny influences the growth strategies of bacterial species after soil wet-up” (Line 272-274); and “These results confirm our second hypothesis that warming and CO₂ enrichment shift the pre-existing growth strategies of certain species” (Line 291-292).

[2] Novelty of this work relative to prior work.

I am loosely familiar with work that is cited in this manuscript (Placella et al) that evaluated a similar concept in grassland soils, published in PNAS 2012. “Rainfall induced CO₂ pulses result from sequential resuscitation of phylogenetically clustered microbial groups”. This research aligns very closely with this study and uses nearly identical methods (qSIP), “taxon specific microbial growth and mortality patterns reveal distinct temporal population responses to rewetting in California grassland soils” Blacewicz et al, 2020. I found myself wondering what sets this work apart from the prior work and the new insights that were provided here. I think the final take-away and why we should care also could be better communicated for relevance to this journals readership.

Response: The major difference between our study and Placella and Blazewicz et al. is

that we conducted our experiment under warming and elevated CO₂ conditions (not done in their studies), which enabled us to disentangle the impact of two factors: phylogeny and environmental acclimation. To the best of our knowledge, we are the first to assess the relative contribution of phylogeny and environmental acclimation to the distribution of microbial growth strategies under ~10-yr climate change scenarios, which is very important in the evolution and ecology field (Maistrenko et al. 2020; Tamames et al. 2016).

We found that phylogeny and environment jointly influence microbial growth strategies, whereby rapid responders were primarily governed by the former and slow responders were primarily governed by the latter. As a result, we have explicitly indicated it as “Our results add new evidence that phylogeny and environmental acclimation are essential to understand the evolution of growth traits of soil microorganisms and their response to climate change” (Line 365-368).

We did use the same method as Blacewicz et al (i.e., qSIP), because qSIP technique can capture the population growth dynamics well (Koch et al. 2018; Sokol et al. 2022). The two articles published in PNAS and ISME J mainly focused on the metabolic or growth response of microbes after rewetting the extremely dry soils (i.e., one ecological process, the “Birch effect”). Different from these two studies, we integrated the ecological responses of soil microbes at different time scales: Firstly, the CO₂ increase in the atmosphere and climate warming are slow, thus soil microorganisms are well conditioned by these global change components (i.e., long-term time scales); Secondly, the response of decomposers to sudden events, e.g. nutrient pulses by rewetting dry soil (i.e., short-term processes), after the long-term acclimation.

References:

- Maistrenko et al. (2020) Disentangling the impact of environmental and phylogenetic constraints on prokaryotic within-species diversity. *The ISME Journal* 14(5), 1247-1259
- Tamames et al. (2016) Quantifying the relative importance of phylogeny and environmental preferences as drivers of gene content in prokaryotic microorganisms. *Frontiers in Microbiology* 7
- Koch et al. (2018) Estimating taxon-specific population dynamics in diverse microbial communities. *Ecosphere* 9, e02090

[3] Experimental design

(1) I have some concerns about the experimental set up. From what I read in figure 1 I did not see a pre-wet treatment (please confirm this), everything was compared to samples collected after the treatment. As cited in the intro, and other papers have all shown, a rapid response is already observed within 1 hour thus growth rate needs to be measured from prewet condition. How long is “immediately”- was the entire experiment set up and then samples removed after? If accurate, address concerns with experimental design, or if I missed it add more clarity to Fig 1.

Response: Sorry for this confusion. In fact, the initial incubation treatment (T = 0 d) is equivalent to the pre-wet treatment. To have a similar operation as other incubation treatments (i.e., incubation for 1, 3, and 6 days), we added water (400 μ l) to the dry soil in the initial incubation treatment. Then, the destructive sampling and soil DNA extraction were conducted (~30 s after water addition). Since the time interval is very short, microbial community profiles in the initial incubation treatment should be the same as that in pre-wet treatment because the prewet soil will also become wet during DNA extraction. Therefore, this could be a minor adjustment but will not affect the experimental results.

To clarify it, we have revised the “initial incubation” to “Prewet soil (0 d)” in Fig. 1 and added the explanatory notes “DNA was extracted from the soils of 0 d incubation treatment immediately after water addition (~30 s interval), representing the prewet treatment” (Line 402-403).

Fig. 1 Field and laboratory experimental design. To examine the effects of CO₂ enrichment (500 ppm CO₂) and warming (+2 °C) for a typical rice-wheat rotation system, an open-air experimental field was started in 2010 (A). Four treatments with twelve 50 m² circular blocks (each treatment with three biological replicates) were set (B). A qSIP incubation experiment was performed to determine the response strategies of active microbes in each climate change treatment (C). Briefly, 2 g air-dried soils with 400 μl of natural abundance water (H₂¹⁶O) or 98 atom% H₂¹⁸O were incubated in the dark at room temperature, with harvests at 0 day, 1 day, 3 days, and 6 days. We calculated the taxon-specific growth rates and determined which of the three pre-defined growth strategies an OTU exhibited in each simulated climate change treatment (D).

I believe this first since not a true control should be T1 (after treatment) as T0 implies before effect and the papers cite a rapid effect following this treatment. This comes important right away as I found myself wondering if on line 122 the differences in richness were due to wetting event or prior historical artifact.

Response: As mentioned above, soil DNA was extracted ~30 s after water addition for the initial incubation treatment. Therefore, it is highly unlikely that microbial community composition was affected by water addition. Because water was added to four climate change treatments simultaneously, we believe that the differences in richness were due to prior historical artifact rather than wetting event.

Note prior study by Blasewitz that authors reference often used these treatments and responder categories.

- a. Prewet {important control
- b. Primary 3 hours
- c. Secondary 24 hours
- d. Tertiary 72 hours
- e. Delayed 168 hours

Response: We did not estimate the growth rates within 3 h incubation. Reviewer #1 pointed out that microbial response to water addition in agricultural soils (used in our study) may be slower and weaker than in extremely dry soils (e.g., soils from Mediterranean climates), whose soil respiration rate and the activity of partial taxa peak within 1h (Placella et al. 2012). Therefore, it could be important to capture microbial growth responses in a short time when studying soils with extremely strong birch effects because of the rapid and strong response of soil microbes. In contrast, agricultural soils that routinely dry and wet back up may experience a minimal birch effect after rewetting (quoted from reviewer #1's comment). His/her thought has been verified by our observation that the soil respiration rate peaked around 9 h after water addition in our experiment, while the maximum respiration rate was lower than the observed values in the study by Placella et al. Accordingly, microbial growth response in our study could be also later than that in the studies shown in Placella et al. 2012 and Blasewicz et al 2020. Thus, the 3 h incubation treatment could be unnecessary for our study.

Our classification criteria of growth strategies could also be valid for identifying rapid responders (e.g., species had maximum growth rates within 3 h), despite the lack of a 3-hour culture treatment. If the growth rate of a species reached a maximum within 3 h and decreased over time, this species can be detected as a rapid responder based on

our pipeline because we estimated the average growth rates of species within 1 day, which included the growth within 3 hours.

References:

Placella et al. (2012) Rainfall-induced carbon dioxide pulses result from sequential resuscitation of phylogenetically clustered microbial groups. *Proceedings of the National Academy of Sciences* 109(27), 10931-10936

Blazewicz et al. (2020) Taxon-specific microbial growth and mortality patterns reveal distinct temporal population responses to rewetting in a California grassland soil. *The ISME Journal* 14, 1520–1532

(2) Prior studies used fresh soils- collected within a week of incubation. However this study used soils that had been stored in the fridge or at 4C. For how long prior to the study being done- I was confused by language at the end of line 435 and the beginning of line 443 (fresh).

Response: We are very sorry for our negligence of the explanation. A portion of the soil samples (~ 60 g) were air dried at room temperature after transport to the laboratory (not stored at 4 °C), used for qSIP incubation. The remaining soil samples were stored in a 4 °C refrigerator.

To clarify it, we have revised the description of the experimental procedure, “approximately 60 g fresh soil of each treatment were sieved (2 mm) and air-dried (24 h at room temperature) immediately after transport to the laboratory” (Line 397-398).

Why were the soils sieved as that disrupts structural features?

Response: The purpose of sieving is to remove the roots and large plant fibers, which can influence microbial activity. Sieving facilitates the homogenization of the soil to avoid differences due to soil heterogeneity. We did it to follow the same experimental procedure in the classic studies that established qSIP techniques (Hungate et al. 2015; Koch et al. 2018).

References:

Hungate et al. (2015) Quantitative microbial ecology through stable isotope probing. *Applied and Environmental Microbiology* 81, 7570-7581

Koch et al. (2018) Estimating taxon-specific population dynamics in diverse microbial communities.
Ecosphere 9, e02090

(3) Line 508: What version of RDP Classifier was used?

Response: We used RDP v16 database for taxonomic annotation. We have added this information in the revision (Line 462).

(4) Line 511: The accession numbers of the deposited 16S rRNA data is stated as “not available until 2023/06/01”. This needs to be updated on the resubmission as we could not download the data.

Response: We have updated the data release date, so the amplicon data are publicly available via the following URL: <https://bigd.big.ac.cn/gsa/browse/CRA006507>.

(5) Line 159 The authors said they used same growth classification as in Blascewicz (which I appreciated linking to prior papers), but I didn't see that manuscript used language of rapid, intermediate, or slow. In fact I couldn't find intermediate in that manuscript – on a quick search.

Response: Our experimental design referred to the study of Blascewicz et al. with modifications (i.e., incubation for 1, 3, and 6 days). However, our approach to dividing the growth strategies was different from that in their study. Since it is a misquote, we have deleted this sentence in the revision.

This becomes an issue and I found myself unsure if their calls for “slow / medium / fast” responders are appropriate as this wasn't entirely easy for me to follow. For example, looking at Figure 2, I can see that abundances for some of the groups are very similar across all days, and so calling an organism as “rapid” because it is ever so slightly higher in sample days 0-1 than in sample days 0-6 seems imprecise.

Response: We apologize for the unclear expression at Line 143 in the prior version, “Taxa with fast growth rates were conserved within several bacterial phyla”. We want to describe which phyla have relatively larger growth rates compared to other phyla

during the incubation, rather than which phyla had rapid growth responses to wet-up. We have revised this sentence to “Taxa with relatively high growth rates belonged to several bacterial phyla (Fig. 2 and Fig. S4), including Actinobacteria ...” (Line 142).

However, when they show Fig 3 and Z-Score those values I can definitely see some distinctions by the growth rate categories. Can the authors clarify how this was done here, as this is important to understand so figures/results can be interpreted.

Response: We defined growth responders at the OTU level. We compared the average growth rates of OTUs at three incubation intervals (i.e., 1, 3, and 6 days) and classified the taxa into rapid, intermediate, and slow growth strategies based on the time point at which the maximum growth rate occurred. To clarify it, we have revised the description of classification approach in the method section, “We compared the average growth rates of taxa at three time intervals (n = 3 in each time interval) and classified the species into rapid, intermediate, and slow growth strategies based on the timing of the maximum growth rate (Fig. 1D): 1) Rapid responders: species had the highest growth rates by 1 day of the incubation; 2) Intermediate responders: species had the highest growth rates at the 3-day incubation; 3) Slow responders: species had the highest growth rates at the 6-day incubation” (Line 489-495).

[4] Language precision

(1) Mainly, the link between microbial taxonomy and microbial metabolism is known to not be straightforward, especially in diverse microbial communities like soils that lack good reference genome databases. I do appreciate that the authors introduce caveats to the end of discussion paragraph 2 very well – and so I am aware they know the limitations of their methods. However, it would be helpful to start framing those caveats earlier within discussion paragraph 1 as well as throughout the results section when the piecrust was introduced. For context, as a reader who does metabolisms in soils I found myself somewhat disappointed when piecrust data was introduced and not properly caveated. Particularly, it might be helpful to introduce briefly at lines 226-236 in the results and lines ~278. These caveats are necessary to not potentially overstate

the author's findings, and to make sure the reader knows the authors understand the very clear limitations of these methods.

<https://microbiomejournal.biomedcentral.com/articles/10.1186/s40168-020-00815-y>

With this approach, we found reasonable performance for human datasets, with the metagenome prediction tools performing better for inference on genes related to “housekeeping” functions. However, their performance degraded sharply outside of human datasets when used for inference.

Also

more: <https://academic.oup.com/gigascience/article/doi/10.1093/gigascience/giab090/6505123>

Response: The Reviewer #2 also criticized that “The Picrust analysis and it's results here and the heat map in Fig 4C are not well integrated with the rest of the manuscript. They feel tacked on and do not meaningfully enhance the manuscript. I suggest they be removed as they detract from the focus of the paper”. Therefore, we have followed the suggestion to remove all of the results from the Picrust analysis.

(2) In the discussion section there must be some additional caveats introduced to some of the claims the authors are making. Some of the language that is used within discussion paragraphs require data that the authors do not have (e.g., whole-genome sequencing or metagenomics), or that are somewhat contradictory to their results (e.g., “we identified major forces of variation” when only 20% of the data is explained by these metrics). As such, softening of the language is recommended- lines are noted below- and these will not adjust the outcomes of this study, but support more precise interpretation.

Response: We have removed a large number of redundant and contradictory sentences to improve readability. We have also involved native English speakers to polish the language. Our revisions are marked in red in the revised manuscript.

Line 274-275: Please be mindful and introduce some caveats to the statements made on

these lines. What else other than phylogeny and “core genes” could possibly drive these differences?

Response: We have added some caveats as “The effects of environmental selection on microbial strategy, however, cannot be ignored (Arnold et al. 2022; Yang et al. 2021). Phenotypic plasticity is crucial in the physiological performance of microorganisms to acclimate to broad and varied environments (Agrawal Anurag 2001). Furthermore, microorganisms can also obtain adaptive genes from other species through horizontal gene transfer (Arnold et al. 2022). These factors could drive these differences of microbial eco-physiological traits” (Line 274-279).

References:

Arnold et al. (2022) Horizontal gene transfer and adaptive evolution in bacteria. *Nature Reviews Microbiology* 20(4), 206-218

Yang et al. (2021) Inferring multilayer interactome networks shaping phenotypic plasticity and evolution. *Nature Communications* 12(1), 5304

Agrawal Anurag (2001) Phenotypic Plasticity in the Interactions and Evolution of Species. *Science* 294(5541), 321-326

Line 330: Consider removing this final sentence The explanation in the upcoming paragraphs is both helpful and needed prior to making the (valid!) claim you are making on line 352.

Response: Done.

Line 366-368: Careful with the phrase “...we identified the major forces...”. Changing this to “we identified some of forces...” is prudent given we do not know all factors. Specifically, there is less than 20% variation explained by these variables in both the rapid and intermediate responders. In other words, 80-89%% of the variation is not explained by these data, and you cannot call them major drivers. Further, please add a strong caveat into this section that speaks to what the other 90% of variability in Rapid and Intermediate responders might relate to, as it is clearly not phylogenetic or environmental acclimation.

Response: We have revised this sentence to “We identified how these two factors shape

microbial growth dynamics” (Line 337-338). We have also added the caveat at the end of this paragraph: “Unexpectedly, less than 20% of variation is explained by these variables for the rapid and intermediate responders, which could reflect other factors shaping the growth strategies. For instance, historical contingencies may affect microbial community composition (Ge et al. 2008), which further affected the current community functions as well as species’ trait expressions. The metabolic processes could be influenced by the co-limitation of multiple resources, which complicates the prediction of microbial physiological performance (Harpole et al. 2011; Ma et al. 2019). Lastly, the undetected changes of soil micro-environment and stochastic processes (e.g., genetic drift), may equally contribute to the observed unexplained components” (Line 347-355).

References:

- Ge et al. (2008) Differences in soil bacterial diversity: driven by contemporary disturbances or historical contingencies? *The ISME Journal* 2, 254-264
- Harpole et al. (2011) Nutrient co-limitation of primary producer communities. *Ecology Letters* 14, 852-862
- Ma et al. (2019) How do soil micro-organisms respond to N, P and NP additions? Application of the ecological framework of (co-)limitation by multiple resources. *Journal of Ecology* 107, 2329-2345

Line 374: Consider softening this language. There are possibilities not accounted for here. “...consisting of numerous proteins, evolved more slowly...” with “...consisting of numerous proteins, usually evolve more slowly...”

Response: We have revised this sentence to “These essential growth-related genes tend to belong to the core genome (Liu et al. 2012), which usually evolved more slowly than the accessory genomes...” (Line 343-345).

Reference:

- Liu et al. (2012) Comparative genomics of *Mycoplasma*: analysis of conserved essential genes and diversity of the pan-genome. *PLoS One* 7, e35698

Line 377: Consider softening this language. Change “...microbial trait of rapid growth

was inherited vertically...” to “...microbial trait of rapid growth was likely inherited vertically...”.

Response: Done. Please check the revised version in Line 346-347.

Line 389-390: The claim that they are governed by specific environments is somewhat misleading. They are primarily governed by environmental variables that may or may not be conserved across an environment. Please rewrite this sentence to reflect that.

Response: We have deleted this paragraph, according to the suggestion of reviewer #1.

Line 392: Consider adding in “...the significance of phylogeny and their respective functional traits...”

Response: Done (Line 357).

Line 393: Per your results, this was not always the case. Please soften this language to reflect your results. “...response to wet-up events are mostly phylogenetically conserved...”

Response: Done (Line 359).

Line 398-399: This claim needs a citation – as you do not have “global” evidence for this. Alternatively, the claim needs to soften the language. “...within a given species locally, and, potentially, globally.”

Response: We have followed the sentence to revise this sentence as “Warming and eCO₂ shift the growth dynamics within a given species locally and, potentially, globally” (Line 363-364).

[5] Specific line comments:

please not just 16S all throughout the manuscript- it should be “16S rRNA gene” to signal primers were used to amplify this gene or similarly “16S rRNA amplicon”.

Response: We have revised “16S” to “16S rRNA gene” or “16S rRNA amplicon” throughout the manuscript.

Line 178 tone down claim- suggesting that “under the conditions evaluated here” vertical inheritance was essential for the distribution of growth rate traits in response to rewetting. I think functional traits is a bit oo broad.

Response: Done (Line 189-190).

Line 60: I believe maybe the intended word was “mediated” instead of “medicated”?

Response: Yes. We have revised it (Line 50).

Line 111: Consider making this title reflect the major result of the section as you did with the rest of the titles. It helps the reader focus on the message.

Response: We have revised this title to “Growth responses of active bacteria as affected by climate change” (Line 114).

Line 133: according to (6). Consider just writing to “according to a previous publication (6)” as the formatting will look better.

Response: Done (Line 132-133).

Line 159: Change “The phylogenetic tree...” to “A phylogenetic tree...” and consider joining the sentence after for clarity. “A phylogenetic tree including all responders ... was constructed, and 3 phylogenetic indices were used to estimate the ...”.

Response: Done (Line 168-171).

Line 170: Consider adding the word additional to “We used two additional indices...”, as you mention above you have 3, and these are the other two.

Response: Done (Line 179).

Lines 226-236: Please add a sentence here regarding the caveat that microbial phylogeny is not always representative of microbial metabolism.

Response: Based on the suggestion from Reviewer #2, we have deleted the results

related to functional prediction from our manuscript.

Line 239: Are you trying to say that all categories, with the exception of intermediate responders in eT, were clustered at phylogenetic branches? If so, please consider writing this sentence like that to enhance clarity.

Response: We have revised this sentence to “All categories, with the exception of intermediate responders in eT, were clustered at phylogenetic branches (NTI > 0, $p < 0.05$, Table 2)” (Line 234-236).

Line 247: Consider changing the word “expresses” to “exhibits”.

Response: We have revised this sentence to “... shifts in species composition and in their response strategy” (Line 245-246).

Line 247: Consider adding the word ultimately to: “...that a species expresses, which ultimately weakened the strength of phylogenetic patterns.” to make your closing statement more powerful.

Response: Done (Line 246). Thanks for this nice suggestion.

Line 281: How many are “most” members? It would be helpful to add in exactly how many.

Response: We have added the specific values in this sentence “...most members of the bacterial classes, e.g., 78% of Bacilli OTUs, 69% of Sphingobacteria OTUs, and 67% of Cytophagia OTUs were classified as rapid responders” (Line 281-283).

Line 315: The tense for this should read: “which is ubiquitous in the microbiome and considerably shapes the structure and functions...”. Style wise I generally avoid the term ubiquitous as it means in everything and since we cannot measure to exhaustion in microbes...but that is my thing I realize.

Response: Agreed. We have deleted this sentence in the revision.

Line 325-328: Please reword this so that it is understood that these findings are from another study. Maybe start the sentence with “A general concept is that these..., and previous works have shown that under resource limitation, per capita resource...”.

Response: Done (Line 294-296).

Line 332: Consider removing “and elevated CO₂ concentration” from this first sentence, as this paragraph is on warming specifically, and the following one is about CO₂ concentration.

Response: Done (Line 302).

Line 345: Consider adding in the removed “and elevated CO₂ concentration” from the comment above here. This then frames this paragraph as the CO₂ paragraph.

Response: Done. We have revised this sentence to “Contrary to the warming effect, an elevated CO₂ concentration increases plant biomass” (Line 312-313).

Line 370: Add the word “and” to “...former, AND slow responders by the latter” for clarity.

Response: Done (Line 340).

Lines 373-378: Careful with the statements in these last few sentences. Some caveats and softening of language here are necessary. The growth-related genes do not need to be located on the core genome, as the referenced paper (27) states. It is possible it is related to environmental adaptations, and some phylogenies are more prone to that than others due to a myriad of factors. Softening the tone with words like “tend to be located” instead of “are located” (line 374).

Response: Done. We have softened this sentence to be “These essential growth-related genes tend to belong to the core genome” (Line 343-344).

Line 541: What happened with the taxa that did not fit into the 3 delimited strategies? Were they removed? Kept? Please elaborate here.

Response: We are very sorry for our unclear writing. Based on our classification method, all detected active species (with growth rates significantly greater than 0) were classified as one of the three growth strategies. We have revised this sentence to “The taxa with growth rates significantly greater than zero can be divided into one of three strategies in each treatment” (Line 495-496).

Figure 6: The labels for “intermediate” say “intermediated” in some instances. Please change to keep consistent.

Response: We have changed them to be “intermediate” throughout the revised manuscript.

Figure S3: Change “birth rates” to “growth rates” to keep consistent with axes label. Also – the legend seems to be missing from this supplementary figure. Please add.

Response: Done.

Also – from the manuscript text: “Taxa with fast growth rates were conserved within several 143 bacterial phyla (Fig. 2 and Fig. S3)”. Could you label these within Figure S3 as you did for Figure 2? What are the smaller bar plots inside some of these bar plots?

Response: We have revised this sentence to “Taxa with relatively high growth rates belonged to several bacterial phyla (Fig. 2 and Fig. S4), including Actinobacteria...” (Line 142-143). We have labeled these phyla with bold in Figure S4.

Since the growth rate differences among treatments were not easily visible for certain taxa (e.g., Deltaproteobacteria and Chloroflexi), we inserted the smaller bar plots with smaller axis orders of magnitude to manifest the differences. We have added this explanation in the legend of Fig. S4.

Fig. S4.

Cumulative population growth rates of dominant phyla or classes. Error bars: standard deviation ($n = 3$). Different letters: significant differences among treatments. For the phyla Deltaproteobacteria and Chloroflexi (relatively low growth rates), the smaller bar plots with smaller axis orders of magnitude were inserted to better highlight the variation of growth rates among treatments. Bold: the top five phyla with relatively high growth rates.

Figure S5: This figure also does not contain a figure legend. Please add.

Response: Done.

REVIEWERS' COMMENTS

Reviewer #1 (Remarks to the Author):

The authors have substantially improved the quality of their manuscript and it is, in my opinion, acceptable for publication. I do have some minor suggestions for edits:

L43 suggest changing “pursuits” to “objectives” and inserting “of microorganisms” after “strategies”

L55 suggest inserting a comma before “featuring” and after “response” and inserting “achieve” before “maximum”

L61 I suggest rewriting the sentence starting with “The phylogeny is crucial....” I find the sentence awkward.

L64 suggest changing “firstly” to “first”

L67 suggest deleting the first sentence of this paragraph.

L89 suggest replacing “trace” with “characterize”

L92 suggest deleting “using”

L115 suggest replacing “enabled to trace the bacterial assimilation of water and” with “was used”

L145 Explain more clearly what the term “growth dynamics” refers to.

L155 delete the first sentence of this paragraph

L207 Use the past tense in this sentence

L242 and L243 delete “the” before “phylogenetic”

L262 -265 delete these sentences

L281 delete “the” before “bacterial”

L293 delete “the” before “negative”

L299 add “values” after “R²”

L312 -318 Use the past tense in these sentences

L345 It is unclear to me what the accessory genomes are

Reviewer #2 (Remarks to the Author):

In general the authors addressed my concerns and the manuscript is significantly improved! I do take issue with the presentation of the new biochemical data (Line 106-113 and Figure S1). It's misleading to present data from two separate soil collections two years apart as occurring at the same time. It should be clear that the nutrient flux data is not on the same samples as the qSIP data.

Line 70 Make sure the citations are still appropriate when you change important language such as ‘adaptation’ and ‘acclimation’. Citation ‘17’ may no longer be appropriate.

Line 188 The word “driven” seems too strong for relatively weak K and lambda values, perhaps change to “influenced by phylogeny”.

Table 2 – the “Phylogenetic relationship” column should be removed. Sufficient information is given in table legend to interpret the results.

244-246 – The phylogenetic organization is only consistently weaker with climate change for the ‘rapid responders’ as currently written it seems like this statement applies to all the groups which is not correct.

Methods – need details on exactly when soil samples were collected for the 18O incubations and need to note that the nutrient flux measurements were taken on samples collected roughly two years later.

Point-to-point Responses to the referees' comments:

Reviewer #1 (Remarks to the Author):

The authors have substantially improved the quality of their manuscript and it is, in my opinion, acceptable for publication. I do have some minor suggestions for edits:

Response: Thanks for the positive comments. We have followed these suggestions, shown in our point-to-point responses below.

L43 suggest changing “pursuits” to “objectives” and inserting “of microorganisms” after “strategies”.

Response: We have replaced this sentence with “This slow acclimation, however, disregards the response of decomposers to sudden events, e.g. nutrient pulses by rewetting dry soil, which strongly impacts biogeochemical processes” according to the editor’s guidance (Line 42-44).

L55 suggest inserting a comma before “featuring” and after “response” and inserting “achieve” before “maximum”.

Response: Done (Line 55-56).

L61 I suggest rewriting the sentence starting with “The phylogeny is crucial....” I find the sentence awkward.

Response: We have revised this sentence to “Microbial phylogeny is directly related to its phenotypic and functional traits” (Line 61-62).

L64 suggest changing “firstly” to “first”

Response: Done (Line 64).

L67 suggest deleting the first sentence of this paragraph.

Response: Done (Line 67).

L89 suggest replacing “trace” with “characterize”

Response: Done (Line 88).

L92 suggest deleting “using”

Response: Done (Line 90).

L115 suggest replacing” enabled to trace the bacterial assimilation of water and” with
“was used”

Response: Done (Line 114).

L145 Explain more clearly what the term “growth dynamics” refers to.

Response: We have revised this sentence to “Some phyla had similar growth dynamics (i.e., changes in growth rates along the incubation time) among climate change scenarios” (Line 144-146).

L155 delete the first sentence of this paragraph

Response: Done (Line 154).

L207 Use the past tense in this sentence

Response: Done (Line 203-205).

L242 and L243 delete “the” before “phylogenetic”

Response: Done (Line 237-238).

L262 -265 delete these sentences

Response: Done (Line 257).

L281 delete “the” before “bacterial”

Response: Done (Line 273).

L293 delete “the” before “negative”

Response: Done (Line 285).

L299 add “values” after “R2”

Response: Done (Line 291).

L312 -318 Use the past tense in these sentences

Response: Done (Line 305-310).

L345 It is unclear to me what the accessory genomes are

Response: We have deleted “accessory genomes” and revised this sentence to “which usually evolved slowly and were unlikely to be transferred among microorganisms” (Line 336-337).

Reviewer #2 (Remarks to the Author):

In general the authors addressed my concerns and the manuscript is significantly improved!

Response: Thanks for the positive comments.

I do take issue with the presentation of the new biochemical data (Line 106-113 and Figure S1). It’s misleading to present data from two separate soil collections two years apart as occurring at the same time. It should be clear that the nutrient flux data is not on the same samples as the qSIP data.

Response: Thanks for this suggestion. We have revised the statement at the beginning of the Results to “Based on additional soil sampling (in July 2022), we confirmed the pulses of available organic substances and nutrients to microbes after rewetting of dry soils” (Line 106-108).

We have added the explanatory notes in the Methods, those are “Soil samples for qSIP incubation were collected in June 2020” (Line 383-384), and “Soil samples for nutrient flux measurements were collected from the free-air CO₂ enrichment and warming experimental station in July 2022” (Line 389-391).

We have added the explanatory note in the figure legend of Supplementary Fig 1, that is “Note that the soil for these analyses was sampled in July 2022”.

Line 70 Make sure the citations are still appropriate when you change important language such as ‘adaptation’ and ‘acclimation’. Citation ‘17’ may no longer be appropriate.

Response: We have replaced the original citation ‘17’ with the article “Thermal acclimation in widespread heterotrophic soil microbes” (Crowther, T. W. and Bradford, M. A, 2013) (<https://doi.org/10.1111/ele.12069>).

Line 188 The word “driven” seems too strong for relatively weak K and lambda values, perhaps change to “influenced by phylogeny”.

Response: We have revised it to “influenced by phylogeny” (Line 184).

Table 2 – the “Phylogenetic relationship” column should be removed. Sufficient information is given in table legend to interpret the results.

Response: We have deleted the “Phylogenetic relationship” column in Table 2.

244-246 – The phylogenetic organization is only consistently weaker with climate change for the ‘rapid responders’ as currently written it seems like this statement applies to all the groups which is not correct.

Response: Thanks. We have revised it to “which ultimately influenced the phylogenetic patterns” (Line 241).

Methods – need details on exactly when soil samples were collected for the 18O incubations and need to note that the nutrient flux measurements were taken on samples collected roughly two years later.

Response: We have added the sentences in the Methods, those are “Soil samples for qSIP incubation were collected in June 2020” (Line 383-384), and “Soil samples for nutrient flux measurements were collected from the free-air CO₂ enrichment and warming experimental station in July 2022” (Line 389-391).